# Local energetic frustration conservation in protein families and superfamilies

Maria I. Freiberger [1,8], Victoria Ruiz-Serra [2,8], Camila Pontes [2], Miguel Romero-Durana [2], Pablo Galaz-Davison [3,4], Cesar A. Ramírez-Sarmiento [3,4], Claudio D. Schuster [5], Marcelo A. Marti [5], Peter G. Wolynes [6], Diego U. Ferreiro[1], R. Gonzalo Parra [2] ✉ & Alfonso Valencia [2,7]

Energetic local frustration offers a biophysical perspective to interpret the effects of sequence variability on protein families. Here we present a methodology to analyze local frustration patterns within protein families and superfamilies that allows us to uncover constraints related to stability and function, and identify differential frustration patterns in families with a common ancestry. We analyze these signals in very well studied protein families such as PDZ, SH3, α and β globins and RAS families. Recent advances in protein structure prediction make it possible to analyze a vast majority of the protein space. An automatic and unsupervised proteome-wide analysis on the SARS-CoV-2 virus demonstrates the potential of our approach to enhance our understanding of the natural phenotypic diversity of protein families beyond single protein instances. We apply our method to modify biophysical properties of natural proteins based on their family properties, as well as perform unsupervised analysis of large datasets to shed light on the physicochemical signatures of poorly characterized proteins such as the ones belonging to emergent pathogens.

Families of proteins originate from a common ancestor and develop over evolutionary timescales through various mechanisms of sequence variability at the domain level[1]. The effects that these variations have on phenotypic traits such as protein stability and biological function may conflict with one another, restricting evolutionary trajectories in the protein sequence space[2]. Multiple sequence alignments (MSAs) of protein families show that there are certain positions under strong evolutionary pressure that have little variability while other positions undergo neutral evolution. The latter ones allow protein

sequences to diffuse in sequence space as long as the mutations preserve the structure of their ground states along with their thermodynamic stability and kinetic accessibility while not compromising function[3]. Superfamilies, families, and subfamilies are terms that have been coined to organize the different levels of sequence, structure, and functional similarity as evolution progresses and phylogenetic trees grow. The MSAs of superfamilies show distinctive patterns of differentially conserved residues that modulate the specificity of biological function within different subfamilies. Methods that represent

[1]Laboratorio de Fisiología de Proteínas, Departamento de Química Biológica – IQUIBICEN/CONICET, Facultad de Ciencias Exactas y Naturales, Universidad de Buenos Aires, Buenos Aires C1428EGA, Argentina. [2]Computational Biology Group, Life Sciences Department, Barcelona Supercomputing Center, Barcelona, Spain. [3]Institute for Biological and Medical Engineering, Schools of Engineering, Medicine, and Biological Sciences, Pontificia Universidad Católica de Chile, Santiago 7820436, Chile. [4]ANID - Millennium Science Initiative Program – Millennium Institute for Integrative Biology (iBio), Santiago 8331150, Chile. [5]Laboratorio de Bioinformática, Departamento de Química Biológica – IQUIBICEN/CONICET, Facultad de Ciencias Exactas y Naturales, Universidad de Buenos Aires, C1428EGA Buenos Aires, Argentina. [6]Center for Theoretical Biological Physics and Department of Chemistry, Rice University, Houston, TX 77005, USA. [7]Catalan Institution for Research and Advanced Studies (ICREA), Barcelona, Spain. [8]These authors contributed equally: Maria I. Freiberger, Victoria Ruiz-Serra. ✉e-mail: gonzalo.parra@bsc.es

proteins, with their sequences as vectors in a generalized sequence space[4] can identify "Specificity Determining Positions" (SDPs)[5], i.e., positions that are differentially conserved within distinct subfamilies. Nevertheless, interpreting SDPs from sequence alone remains challenging. Energy-based structural approaches become useful, since sequence diverges faster than structure[6], and structural comparison has facilitated to group evolutionarily related protein families into superfamilies, even in the absence of detectable sequence similarity[7].

Sequence variations can be linked with their structural, functional, and dynamic consequences using the concept of local energetic frustration[8], derived from the energy landscape theory of protein folding. According to the "Minimal Frustration Principle", possible strong, energetic conflicts between different residues are minimized in the native states of foldable proteins, unlike random heteropolymers. Nevertheless, some conflicts may have been positively selected by evolution due to functional requirements. These conflicting signals have been shown to be enriched around residues that are associated with different functional aspects of proteins, such as the binding to small ligands or cofactors as well as protein-protein interactions[8], allosterism[9], catalytic sites[10], disorder/order transitions[11] or the existence of fuzzy regions[12]. Because amino acids at those positions are selected for functional reasons, they can often lead to more rugged energy landscapes, resulting in a trade-off between molecular function and local stability[13]. Many proteins have been shown to have reduced activity when stabilizing mutations are introduced at functional sites, showing the delicate equilibrium between stability and function and the functional importance of local frustration[14,15]. Such local frustration can be quantified with the frustratometer algorithm[16,17] by comparing the native energy of individual residues or pairs of residues to a random background energy distribution that results from generating a set of decoys.

Here we explore the concept of local energetic frustration in the evolutionary context of protein families, going beyond single protein instances. The analysis of the conservation of local frustration levels permits the identification of common and differential patterns among evolutionary-related proteins. Our rationale is that the conservation of minimally frustrated interactions within a protein family over extended evolutionary timescales implies their crucial role in foldability or local stability. Furthermore, the conservation of highly frustrated interactions suggests that such local, unfavorable energetic conditions are required by specific functional requirements that have persisted over the evolutionary history of the family. We show the usefulness of the evolutionary analysis of frustration in protein families to address several questions using both experimental structures as well as structural models. We used such an analysis to retrieve experimental measurements of physicochemical changes in proteins belonging to the PDZ, SH3, and KRAS families; and study the functional and structural divergence of related protein families such as α and β globins and the RAS subfamilies. Moreover, we illustrate the general applicability of these ideas by developing an unsupervised strategy to rapidly uncover sequence and energetic constraints in large datasets like the entire SARS-CoV-2 proteome. Finally, we provide an example of how such strategy can guide attempts to modify the biophysical behavior of the metamorphic RfaH protein based on its family frustration conservation patterns.

In this work, we show how the analysis of frustration in an evolutionary context can provide valuable insights into the conservation and divergence of structural and functional properties within protein families. The recent advances in protein structure prediction[18,19] that have made high-quality structural models available to most known proteins have unlocked the applicability of our methodology to most protein families that are deposited in bioinformatic databases.

## Results

### The conservation of frustration in protein families
We have developed a methodology, implemented in a tool that we name FrustraEvo, to measure the conservation of local energetic

frustration over aligned residues or contacts in a protein family. Local energetic frustration measures how well optimized for folding the energy of a given residue-residue interaction is in comparison to the random interactions that would occur within the polypeptidic chain in non-native conformations. See the Methods section for more complete details on how local frustration is calculated.

Local frustration conservation analysis can be carried out using any of the 3 Frustration Indexes (FIs). For simplicity, we next explain the methodology based on the Single Residue FI (SRFI) although the same analysis can be generalized to the pairwise contacts FIs. Given an MSA and the corresponding structures for each sequence contained in it, we can compare the local frustration values from all of the structures at each aligned residue within the MSA. In Fig. 1a, we show this mapping for a region of the α globin family. Some columns in the MSA show more frustration conservation compared to others. We can quantify the evolutionary significance of such conservation by calculating the Information Content (IC) for each MSA column (see Methods), based on the distribution of local frustration states (FrustIC). The more conserved the frustration state is at a given MSA position, the higher its FrustIC will be, and conversely, the lower the FrustIC is the more similar the distribution of frustration states at that MSA position is to the background distribution. In Fig. 1b, we show the SRFI for the human α globin protein (FrustratometeR results[17]) as well as the FrustIC values computed from the α globins MSA (FrustraEvo), mapped on top of the human α globin structure. While some frustration states at the individual protein are consistent with the ones being conserved at the family level, others are not, reflecting protein-specific, perhaps evolutionarily divergent characteristics (Fig. 1b). Similarly to conservation states, we calculate SeqIC, which measures the conservation of amino acid identities within the MSA columns. A schematic view of FrustraEvo's workflow is shown in Fig. 1c (see Methods for details).

### Evolutionary analysis of frustration unveils stability constraints within protein folds
To investigate the evolutionary role of frustration in foldability, stability, and function, we analyze three specific family cases for which double-deep protein fragment complementation (ddPCA) experiments have been performed and are available. This experimental method has been used to quantify the effects of amino acid variation on protein stability (abundance) and function (binding)[20,21]. For each position of the C-terminal SH3 domain of the human growth factor receptor-bound protein 2 (GRB2-SH3), the third PDZ domain of the adaptor protein PSD95 (PSD95-PDZ3) and the GTPase KRas (KRAS), the experiments have produced two scores (ddPCA phenotype scores), corresponding to the effect of variations on stability and function.

We automatically retrieved homologous proteins for the 3 studied proteins, although for KRAS we also analyzed a highly curated dataset from Rojas et al.[22] (see Methods). We analyzed the relationship between the calculated SeqIC and FrustIC (Fig. 2a and Supplementary Fig. 1a, b) values for individual MSA positions with the ddPCA experimental abundance scores derived for the individual proteins. Surprisingly, even when ddPCA measures global properties of the studied systems and frustration a local one, we found that FrustIC is a good predictor of the experimental ddPCA phenotype scores for stability and function for SH3 ($r = -0.79$, $p$ value $= 9.5\text{e-}05$) and PDZ ($r = -0.82$, $p$ value $= 7.6\text{e-}08$) (Fig. 2c, e and Supplementary Fig. 2a, d). The SeqIC score is also correlated with the ddPCA phenotypes although with a slightly lower Pearson correlation coefficient ($r = -0.63$, $p$ value $= 2.5\text{e-}07$ and $r = -0.69$, $p$ value$=6\text{e-}13$, respectively) (Fig. 2b, d, Supplementary Fig. 2b, e). Furthermore, SeqIC and FrustIC are correlated to each other in both proteins (Supplementary Fig. 2c, f). On the other hand, KRAS shows no significant correlation between the ddPCA scores and SeqIC (Fig. 2f) but has a significant and moderate correlation with FrustIC for the minimally frustrated and conserved residues ($r = -0.47$, $p$ value $= 0.00012$) (Fig. 2g). The correlation between SeqIC and

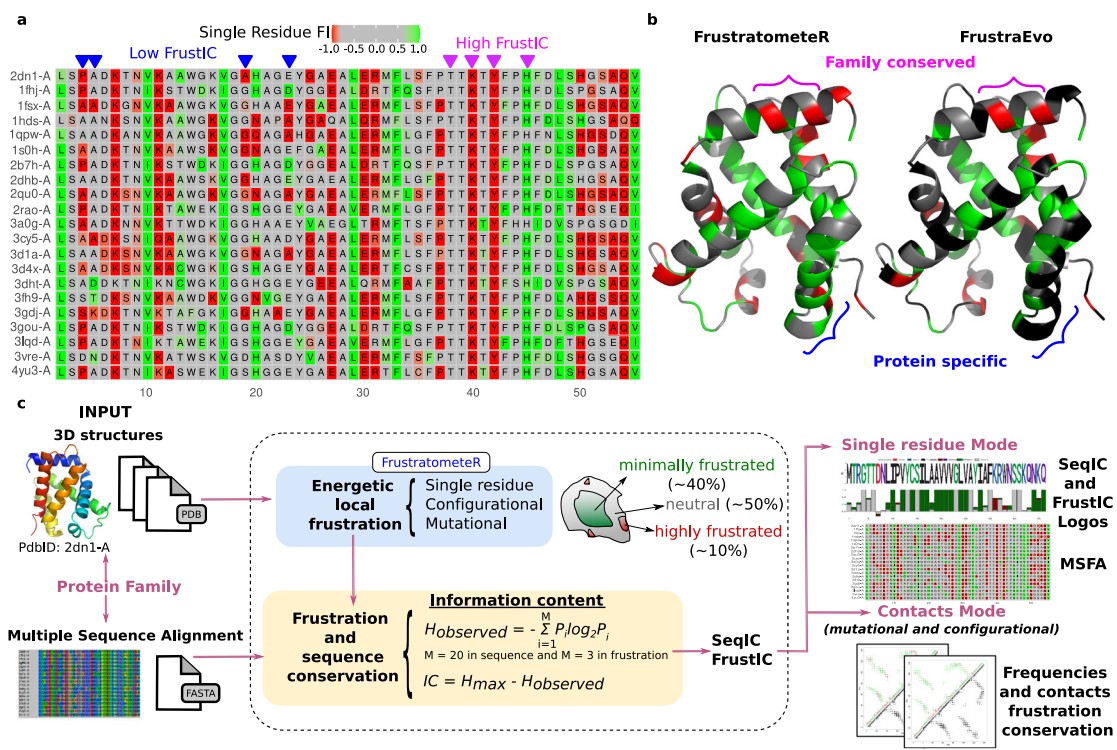

**Fig. 1 | Analysis of frustration in protein families. a** Multiple Sequence Frustration Alignment (MSFA) that consists of the SRFI computed from individual protein structures mapped into the MSA (see Methods). Residues in the MSA are colored according to their SRFI in the corresponding structures. Magenta inverted triangles mark frustrationally conserved residues (high FrustIC), and blue ones mark non-frustrationally conserved residues (low FrustIC). Minimally frustrated residues are colored in shades of green, neutral in gray and highly frustrated in red. **b** Comparison between SRFI values as calculated by FrustratometeR (left) and the conservation of frustration states based on their FrustIC values as calculated by FrustraEvo (right) visualized in the same structure (human α globin, PDB 2DN1, chain A). Residues are colored according to their frustration states in the FrustratometeR representation. Residues with FrstIC > 0.5 are colored according to their most informative frustration state in the FrustraEvo representation, while residues with FrstIC ≤ 0.5 are colored in black. **c** Overview of the FrustraEvo workflow to analyze a single protein family.

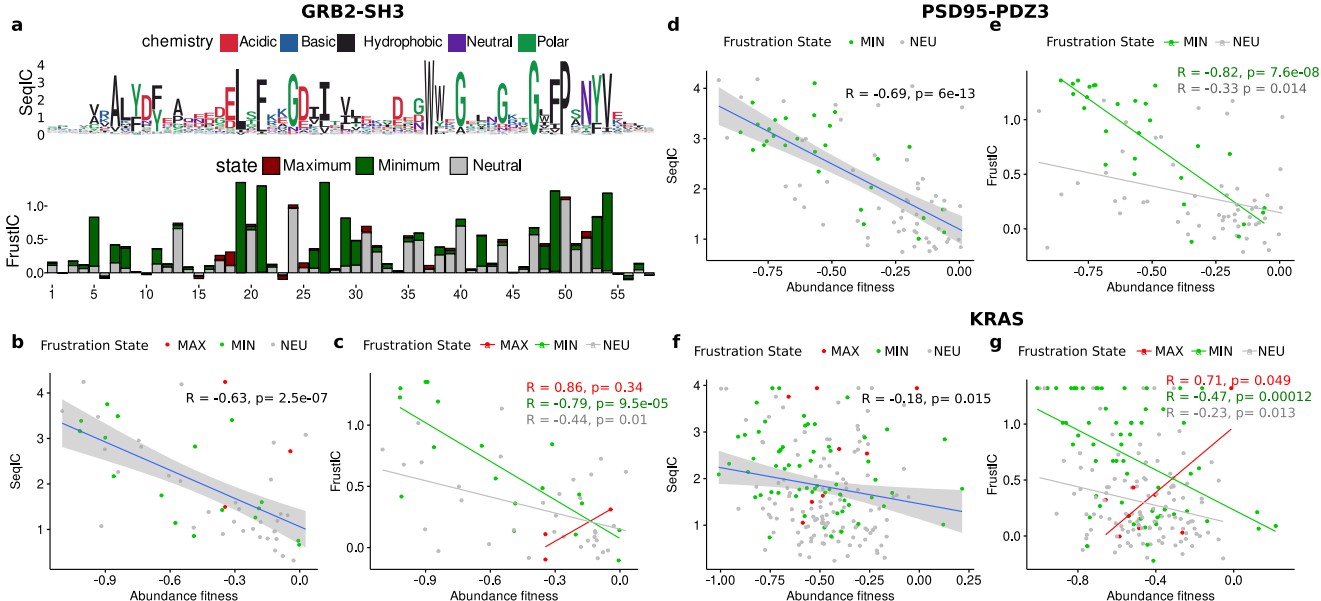

**Fig. 2 | FrustIC correlates with experimentally measured protein stability changes. a** Sequence and Frustration logo plots showing SeqIC and FrustIC values per MSA column, respectively for GRB2-SH3. The numbering of the plot corresponds to the sequence of reference (chain A from PDB 2VWF in the case of GRB2-SH3). Positions containing a gap in the sequence of reference are not considered in the plot. **b–f** Pearson correlation between ddPCA abundance scores vs SeqIC and FrustIC for GRB2-SH3 (**b, c**), PSD95-PDZ3 (**d, e**) and KRAS (**f, g**). P value corresponds to a two-sided test. Error bands in the correlation plots correspond to a 95% confidence interval. Source data are provided as a Source Data file.

FrustIC is also weaker (Supplementary Fig. 2i). Most residues with FrustIC values close to the theoretical maximum ($\log_2(3) = 1.58$, see Methods) are located in the hydrophobic core of the protein (Supplementary Fig. 1c). Because they can vary between the hydrophobic types of amino acids without affecting stability too much, their FrustIC values correlate better with abundance ddPCA score than the SeqIC values (Supplementary Fig. 1b). Relationships between SeqIC, FrustIC and the ddPCA binding score are shown in Supplementary Fig. 2.

Both SH3 (Fig. 2a) and PDZ domains (Supplementary Fig. 1a) do not have highly frustrated conserved positions (FrstIC > 0.5). As these are protein-protein interactors with no localized function, highly frustrated residues are not aligned in the family MSAs and hence their signal gets averaged out. In contrast, KRAS has one highly frustrated conserved position (FrustIC > 0.5, Supplementary Fig. 1b), K117 (KRAS numbering), which is one of the seven conserved residues that interact with the nucleotide substrate[23]. Remarkably, K117 has a very high ddPCA value. This means that most mutations in that residue improve the protein foldability and stability, reinforcing that functional signals often conflict with foldability and stability ones at that locus. As explained in the methods, substrates are invisible to the frustration calculations (as they are not parametrized in the energy function of the algorithm), and therefore the energy at the sites where they bind to the proteins is not compensated, resulting in the presence of highly frustrated interactions that otherwise would be minimally frustrated in their presence. It is interesting, however, that from all the residues that are in contact with the substrate, our analysis highlights K117 as the most important one, as known from biochemical assays.

The KRAS results that have been discussed correspond to the curated dataset from Rojas et al. We further repeated the analysis using the automatic retrieval of homologous proteins (see Methods) and found that the same trends were recovered although the correlation between FrustIC and the abundance ddPCA score is weaker (Supplementary Fig. 3b). This might be because the homologous relationship in the Rojas et al. dataset (36 proteins) was defined by phylogenetic studies while in the other (1354 proteins) it was assumed after performing Blast and simple quality filters (see Methods). This highlights that although our strategy can retrieve meaningful results from automatically generated datasets, highly curated ones, with a fine-tuned definition of the family, will perform better.

The correlation values between ddPCA scores and FrustIC are not as high as they could be (-0.8 for PDZ and SH3 and 0.47 for KRAS), meaning that other factors beyond local frustration need to be taken into consideration to predict global stability as captured by ddPCA. However, the correlation is good enough and significant to showcase the usefulness of FrustIC as a decent in silico prediction of stability related to specific residues when no experimental data is available.

Frustration signals conserved at the family level have appeared in the ancestor of the family and have been maintained invariantly since then for foldability or stability (minimally frustrated levels) or functional requirements (highly frustrated levels). Conversely, if a position shows large variability, it suggests that no strong constraints exist in that position, and therefore, sequence identity can drift. In the case of protein interactors, like the SH3 and PDZ domains, the binding interfaces have adapted to the binding of different partners or ligands by each family member, rendering a lack of conservation of highly frustrated positions. For the contrary, what remains conserved across the family are the common folding and stability properties, that are detected as minimally frustrated positions. We observed a similar situation for members of the Ankyrin repeat protein family[24] that mainly function as protein-protein interactors. In contrast, KRAS has a localized function to bind a nucleotide and cofactors, probably involving other mechanisms such as allosteric regulation. Because these functional signals are localized consistently in specific sets of residues within the MSA, their conflictive signals affect the overall stability of the protein, being a cause for the lower FrustIC-ddPCA correlation. We have also reported similar trends for two enzymatic protein families, i.e., Beta Lactamases and Aldolases[10].

## Differential frustration conservation patterns reveal family-specific functional adaptations within protein superfamilies

We further investigated the link between sequence divergence and the divergence of local energetic frustration by analyzing variability among evolutionarily related, but distinct, protein families. By comparing frustration conservation patterns between evolutionary-related families, differences can be interpreted as the result of functional adaptations in each of them since diverging from their common ancestor. We analyzed the common and differential frustration conservation patterns for two very well-studied examples; first comparing the α and β globins, which are parts of the hemoglobin biological unit, and then examining the human RAS superfamily. The globins are relatively closely related to each other, while the RAS superfamily has experienced wide sequence divergence.

The α and β globin subfamilies have a common origin, but despite their very similar structures, they have different and well-studied functions, as part of the hemoglobin α2β2 tetramer[25]. We used a non-redundant set of experimental structures that correspond to 21 mammalian hemoglobins (see Methods). Figure 3a shows the frustration conservation patterns, based on the SRFI, for the α and β families grouped into a single dataset (α/β dataset). Frustration level is mostly conserved (FrustIC > 0.5) at minimally frustrated positions ($n = 35$, mean FrustIC = 1.02) and at neutral positions ($n = 34$, mean FrustIC = 0.85). Only 3 positions are highly frustrated (mean FrustIC = 0.72).

Some positions in the α/β MSA show changes in their amino acid identity that result in different frustration states being conserved between the α and β families. For example, position 39 (numbering corresponds to the reference structure, PDB ID: 2DN1, which is position 40 in the uniprot entry) in the α/β MSA, corresponds to a highly frustrated Lys in the α family (Lys40α) (Fig. 3b) but to a neutral Gln in the β family (Gln39β) (Fig. 3c). This suggests that a functional adaptation occurred at that position after the divergence of the two families with more functional constraints in α globins compared to β globins. To further study such types of positions we have used the S3Det software[5] to detect SDPs between the two globin families and to analyze their relationship with frustration conservation. There are 15 SDPs between α and β globins of which only 6 differ in their frustration states (Supplementary Table 1). Interestingly, such energetics are not trivially explainable from sequence identity. Some SDPs maintain consistent energetic levels and conservation despite changing identity. Position 32 shows minimal frustration both in α globins (F) and in β globins (L). Other SDPs differentiate clusters based on frustration levels, like position 57 being neutrally conserved (S) in α globins and maximally frustrated (N) in β globins. However, some SDPs of the same amino acid type, do not exhibit consistent frustration conservation, such as position 140 being minimally frustrated (V) in α globins and neutrally frustrated (A) in β globins.

As mentioned earlier, highly frustrated interactions are usually suggestive of local, functional requirements. In total, there are 12 highly frustrated positions in α globins (mean FrustIC = 0.87, Fig. 3b) and 8 in β globins (mean FrustIC = 0.88, Fig. 3c) with only two residues (Q54α, K59α and Y140α, Y145β) being common to both families. This points out at differential functional adaptations that have happened independently within each family after diverging from their common ancestor. Several of these loci correspond to residues involved with the asymmetric interactions of each subunit within the tetrameric structure of hemoglobin, i.e., K39α, and Y42α and W37β, N57β, E101β and N108β. Other highly frustrated residues correspond to the differential function and structural details of each subunit type, e.g., K99α and S124α interact with the αHb-stabilizing protein (AHSP), a

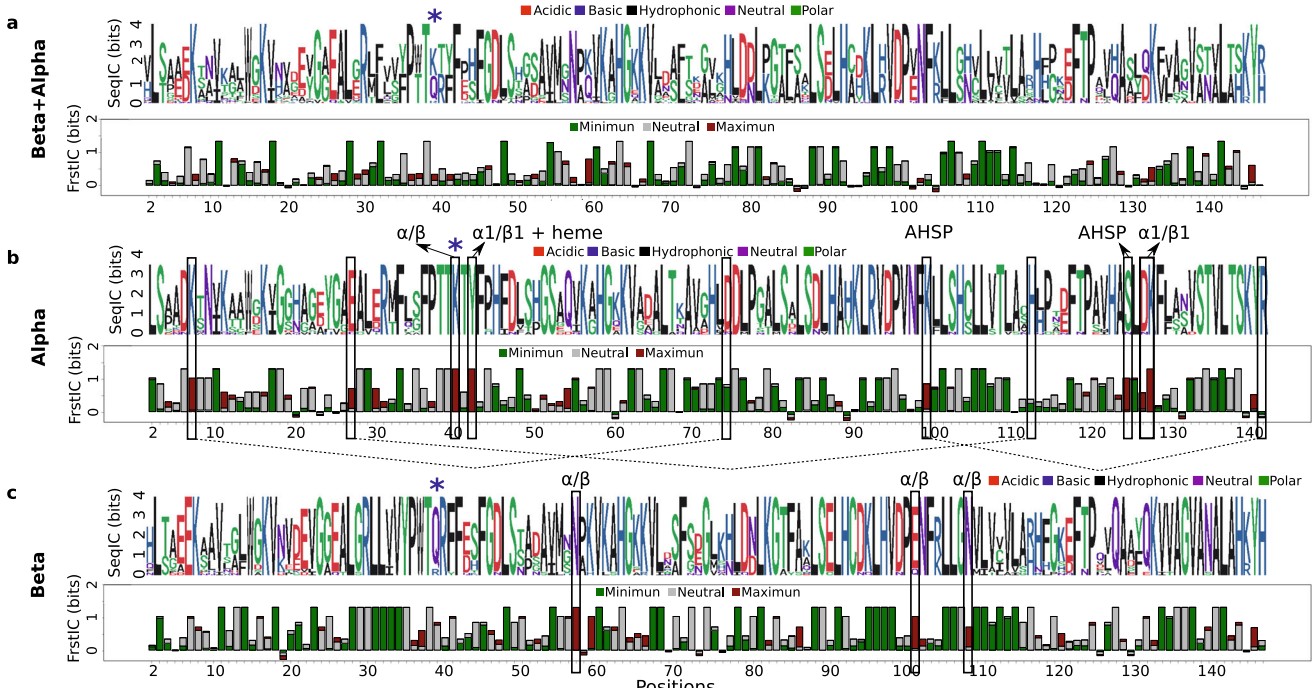

**Fig. 3 | Differential frustration conservation patterns unmask functional constraints in the hemoglobin subunits.** FrustraEvo results based on the SRFI for **a** α/β-globins, **b** only α, and **c** only β. Rectangles denote functionally relevant positions explained in more detail in Supplementary Table 2. In blue asterisks, we marked position 39 in the α/β MSA, which corresponds to a highly frustrated Lys40α but to a neutral Gln39β. The reference structure for this analysis corresponds to the Human Hemoglobin PDB 2DN1. Source data are provided as a Source Data file.

chaperone that prevents α-globin toxicity when isolated[26,27] (Supplementary Table 2), a function that is not shared with the β subunits. In addition, K7α, E27α, and E30α form intra-subunit salt bridges (as shown in Fig. 3)[28] that are critical for allostery and the Bohr effect as explained by Perutz[29] (Fig. 3b, Supplementary Table 2). The Bohr effect is tightly related to allosteric tensed (T) or relaxed (R) states equilibria, which shift towards the T state due to reduced pH and higher $CO_2$ partial pressure, resulting in better oxygen release in the tissues. Therefore it is tightly related to the α/β interface. Indeed, several of the highly frustrated, high frustIC residues identified in both subunits are key for the switch, such as K40α, Y42α and W37β and E101β. Interestingly, other residues showing high SRFI and FrustIC, such as K127α or N57β are located at positions where mutants have been shown to subtly change oxygen affinity (see Supplementary Table 2), thus suggesting they are also involved in this allosteric equilibrium.

Our approach can be applied to much more complex examples. We have analyzed the human RAS superfamily, composed of 5 subfamilies (RAS, RHO, RAB, ARF, and RAN) that have undergone extensive sequence and functional diversification. All the subfamilies share a common structure and enzymatic activity related to GTP hydrolysis. The GTP-binding site, consisting of five motifs (G1–G5), plays a crucial role in GTPase activity. Analysis using the SRFI shows little conservation for the G1-G5 motifs (Supplementary Fig. 4) with only a few positions with conserved minimal or neutral frustration values (FrstIC > 0.5) across all families (Supplementary Fig. 5a). Instead, we explored frustration conservation at the level of residue-residue contacts based on the mutational FI. A network of highly frustrated interactions mainly involving residues that interact with the substrate (e.g., Lys16 (G1), Asp57 (G3), and Lys117 (G4)) is conserved in all four subfamilies. These residues appear as highly frustrated because the substrate is not parameterized in the FrustratometeR energy function and is invisible during the energetic assessment. Therefore, the energy is not compensated as it would be if interactions with the substrate were explicitly considered (see Methods). In an uncharacterized family, the presence of such a conserved network would point out

functional requirements that could be further explored. Interestingly, many SDPs are part of this network and it can be seen that the changes in amino acid identities among the subfamilies are translated into changes in frustration levels. As an example, SDP 83 (RAS numbering) is a highly conserved Asp with highly frustrated interactions (Supplementary Fig. 5b) in ARF and RAB. In contrast, it is a conserved Ser in RHO that establishes minimally frustrated interactions. The identity of the residue is not conserved in RAS, having a mixture of both highly frustrated and minimally frustrated interactions. A more detailed analysis of the Ras superfamily is explained in Supplementary Note 1.

Frustration conservation analysis allows one to interpret sequence diversity among evolutionarily related families and to link this diversity with functional adaptations within divergent protein families. In some cases, as for the globins, frustration conservation analysis using the SRFI is sufficient to uncover these stability and functional signals, while in more divergent examples, conservation analysis using the pairwise contacts FIs is more enlightening.

## Large-scale application of frustration conservation analysis in coronaviruses

A valuable use of analyzing the conservation of frustration is to provide insights about proteins that are still poorly characterized in the laboratory, such as those from emergent pathogens. To illustrate this application, we have automatized the steps of generating MSAs, clustering the resulting subfamilies based on the SDP methodology[5] and have used structural models predicted by AlphaFold2 and combined these steps with the use of FrustraEvo using the SRFI. We applied this workflow to the full SARS-CoV-2 proteome in the context of the entire Coronaviruses phylogeny (see Methods; data available in Zenodo, see Data availability). Although there is a significant correlation between SeqIC and FrustIC (Fig. 4a; $r = 0.69$, $p$ value = 2.6e-14), many proteins deviate from the expected values due to factors such as the presence of disordered regions or the diversity of the MSAs (Supplementary Fig. 9). In addition, different families within SARS-CoV-2 show large

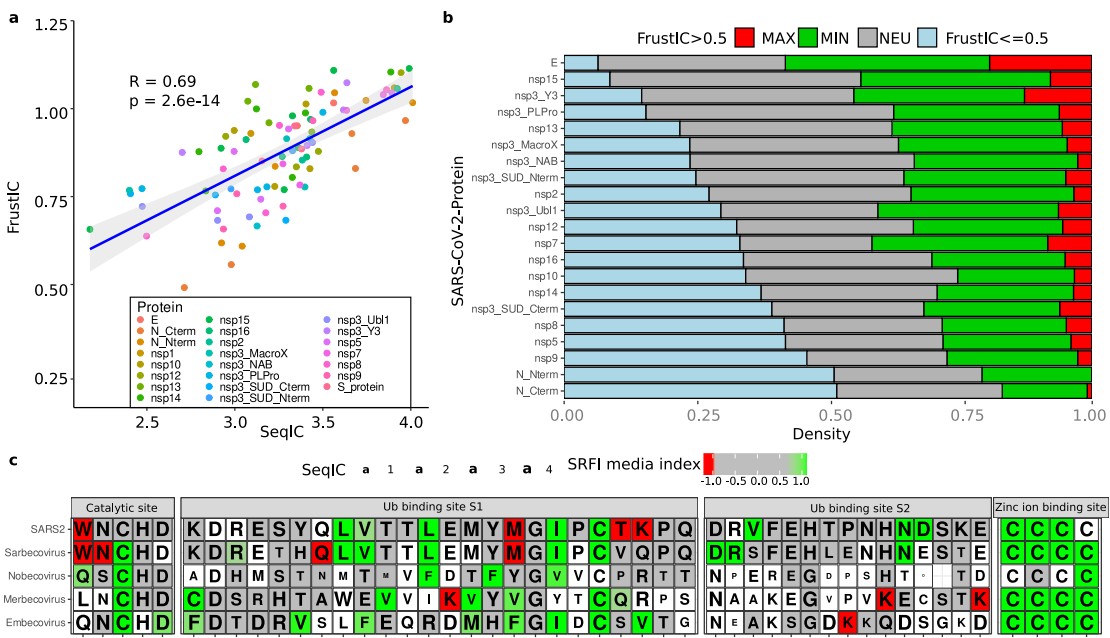

**Fig. 4 | Large-scale application of frustration conservation analysis in coronaviruses. a** Pearson correlation plot showing mean FrustIC vs mean SeqIC per S3Det cluster computed for Coronavirus proteins (see Methods). *P* value corresponds to a two-sided test. Error bands in the correlation plots correspond to a 95% confidence interval. **b** Distribution of frustrationally conserved residues (FrustIC > 0.5) for each S3Det cluster containing the corresponding SARS-CoV-2 protein. We considered frustration conservation when FrustIC > 0.5. The proportion of each protein is normalized by its length (Supplemental Table 3). **c** MSFA showing FrustraEvo results for selected functional domains in PLPro. Cells that are colored correspond to FrustIC > 0.5, while white cells mean that FrustIC ≤ 0.5. Color of the cells represents the median SRFI value computed with FrustratometeR (see methods for frustration states definitions). The amino acid identities correspond to the consensus sequence, and the size of the letter is proportional to SeqIC. Source data are provided as a Source Data file.

differences in terms of the proportion of positions that are energetically constrained suggesting differential evolutionary pressure on them (Fig. 4b). When comparing protein families across different Coronaviruses we also observed a large variability between different subfamilies. Some positions in some proteins are conserved across the entire phylogenetic tree, while others have energetic signatures that are specific to some subfamilies like the Sarbecoviruses or to specific viruses like the SARS-CoV-2. See Supplementary Note 2 for a more detailed analysis of the results for the 29 proteins or protein domains contained within the SARS-CoV-2 full proteome and comparison across the Coronavirus phylogeny.

To better exemplify the impact of our approach to provide functional insights about specific proteins, we analyzed in more detail one of the viral domains of higher functional relevance: the Papain-like Protease (PLPro) protease domain. PLPro catalyzes the proteolysis of the viral polyproteins[30]. Moreover, PLPro interacts with at least two host proteins, ubiquitin-like interferon-stimulated gene 15 protein (ISG15) and ubiquitin (Ub), to evade or at least hamper the host immune response[31]. SARS-CoV-2 PLPro homologous proteins were automatically divided into 4 subfamilies that reflect the Betacoronavirus subgenera classification, i.e., Sarbecovirus (*n* = 31), Nobecovirus (*n* = 11), Merbecovirus (*n* = 35) and Embecovirus (*n* = 45) (Supplemental Table S4). Additionally, we have manually analyzed a fifth group that only contains experimental SARS-CoV-2 PLPro structures (*n* = 29) to quantify frustration conservation specific to this virus. We compared frustration conservation between the 4 PLPro subfamilies to disclose functional diversity related to differential infectivity or virulence (Fig. 4c). At the catalytic site, the SDP Trp106, which facilitates catalysis[31] by stabilizing the catalytic triad, is conserved both in sequence and in its highly frustrated state only in the Sarbecoviruses group. In that same position, Merbecoviruses have a conserved Leu that is not energetically conserved, which is reported to make catalysis less efficient[32]. When Leu 106 is replaced by a Trp in MERS, catalysis is

enhanced, suggesting that increased local frustration may be related to the improvement of the catalytic function. In contrast, the catalytic residue Cys111 is minimally frustrated and conserved in all subfamilies, reflecting the functional importance of local stability at that position to the full phylogeny. In the SARS-CoV-2 set of structures, this position appears neutral, due to the occurrence of a subgroup that contains the Cys111Ser mutation, which introduces local instabilities. Likewise, the four cysteines (Cys189, Cys192, Cys224, and Cys226) that coordinate the binding of an ion of Zinc, indispensable for the functioning of the protein[33], are all minimally frustrated and conserved in most of the coronavirus subfamilies, suggesting a strong stability requirement in that region (Fig. 4c).

Additionally, the PLPro binding sites to ISG15 and Ub host proteins (S1 and S2, respectively), are differentially conserved between the four Betacoronavirus subfamilies (Fig. 4c). The SARS-CoV-2 S1 site contains more highly frustrated residues while the S2 site contains more minimally frustrated residues than the other viruses. This could explain the differential preference that PLPpro has for binding to ISG15 or Ub in SARS-CoV and SARS-CoV-2[34]. For instance, some positions within the S1 region are highly frustrated only at the SARS-CoV-2 level. Positions 225 and 232 (SARS-CoV-2 numbering) (Fig. 4c) correspond to neutral Val and Gln in Sarbecovirus but to highly frustrated Thr and Lys in SARS-CoV-2. It has been shown that these changes affect Ub association, explaining the differential activity on Ub substrates but not on ISG15[31]. Thr225 is only present in SARS-CoV-2 and in RaTG13 (Supplementary Fig. 11), the latter being a likely bat progenitor of the COVID-19 virus[35]. Moreover, the bat-derived viral strains, Rc-o319, and bat-SL-CoVZXC21, contain a Met in that position that is even more frustrated (Supplementary Fig. 11). This may point to a position of concern for novel human-infecting variants that could acquire this change of identity. Lys232 is unique to SARS-CoV-2 within the Sarbecovirus family, suggesting a recent gain of function event.

## Perturbing conserved frustration patterns to engineer conformational changes in a metamorphic protein

The C-terminal domain (CTD) of RfaH, a bacterial elongation factor, undergoes a large, reversible structural rearrangement from an α-hairpin (αCTD) into a β-barrel (βCTD) upon interacting with RNA polymerase and specific DNA elements called *ops*[36]. In the absence of *ops* elements, RfaH αCTD is autoinhibited by being bound to the N-terminal domain (NTD) hence, preventing the correct interactions with the RNA polymerase. On the other hand, NusG, a non-metamorphic paralog from which RfaH is believed to have originated via gene duplication divergence, only exists in its βCTD fold[37]. As the RfaH capacity to undergo metamorphosis seems to have occurred after its divergence from NusG, we used FrustraEvo to find the constraints that are present in the RfaH subfamily when in its autoinhibited αCTD conformation and explored which perturbations would facilitate its transition into the βCTD fold.

We retrieved a set of non-redundant, evolutionarily related RfaH protein sequences (see Methods), predicted their structures with AlphaFold2 (see Methods), and computed their frustration conservation patterns using both the SRFI and the mutational and configurational pairwise contacts FIs. Based on the SRFI, only two residues are consistently frustrated across all family members (predominantly red positions in Supplementary Fig. 12 with FrustIC > 0.5). Their structural location, far away from the metamorphic domain (residues 110-162), suggests that they do not play a role in RfaH fold-switching. In contrast, based on the configurational and mutational FIs, we found a group of highly conserved and minimally frustrated contacts located at the interdomain interface between the αCTD and NTD domains (Fig. 5a). This is consistent with the stabilization via interdomain interactions[38,39] which trigger the RfaH fold-switch towards the βCTD conformation[40,41] when disrupted. We selected 9 interdomain residues (L6, F51, L96, F126, I129, L141, L142, L145, I146) according to their contribution to the interdomain interface stabilization between RfaH NTD and αCTD (see Methods) and used FrustratometeR[17] to predict the changes in frustration when individually mutating them to all other 19 amino acids

(Supplementary Fig. 13). Figure 5b illustrates how the local frustration changes upon mutation between F51 and all the residues with which it interacts. Most of the 21 contacts formed by F51 (Fig. 5b, blue letters) are minimally frustrated, with the exception of 5 that are neutral. Overall, some mutations yield similar frustration values across all contacts (e.g., F51M), while others switch from minimally frustrated interactions to neutral or highly frustrated (e.g., F51K). The same effect is observed when repeating the analysis for the remaining 8 interdomain residues (Supplementary Fig. 13), leading to the identification of two types of possible mutations: 1) "Similar Frustration Mutations" (SFMs), which would maintain the stabilizing nature of the native amino acid identities (L6I, F51M, L96W, F126W, I129V, L141V, L142V, L145M, I146V) and 2) "Highly Frustrated Mutations" (HFMs), which would maximize the local frustration index with their neighboring residues (L6D, F51K, L96K, F126N, I129E, L141D, L142K, L145E, I146D). We generated two *E. coli* RfaH mutant sequences containing all SFMs or HFMs and predicted their structures with AlphaFold2 (see Methods). Structures with SFMs show a similar structure to the wild-type with an αCTD conformation (Fig. 5c), while the ones containing the set of HFMs show a conformational change similar to βCTD (Fig. 5d). The latter suggests that evolution may have positively selected such favorable interdomain interactions that would in turn favor the αCTD conformation foldability over the βCTD one. In turn, this would have enabled metamorphosis as a regulatory mechanism where interactions with the *ops* elements, with RFaH in its inhibited conformation, would have a destabilizing effect on the interface between NTD and αCTD triggering its metamorphosis towards the βCTD conformation to be able to interact with the RNA.

We also investigated which sequence changes were introduced by evolution between RfaH and NusG, its non-metamorphic homolog. Six of the 9 interdomain residues change their identity in the RfaH/NusG sequence alignment (i.e., NusG-like mutations: L6V, I129V, L141V, L145I, I146F and L142S, Supplementary Fig. 14a). We introduced these mutations into the RfaH sequence and found that 4 out of 5 of the top AlphaFold2 structure predictions display a βCTD-like fold

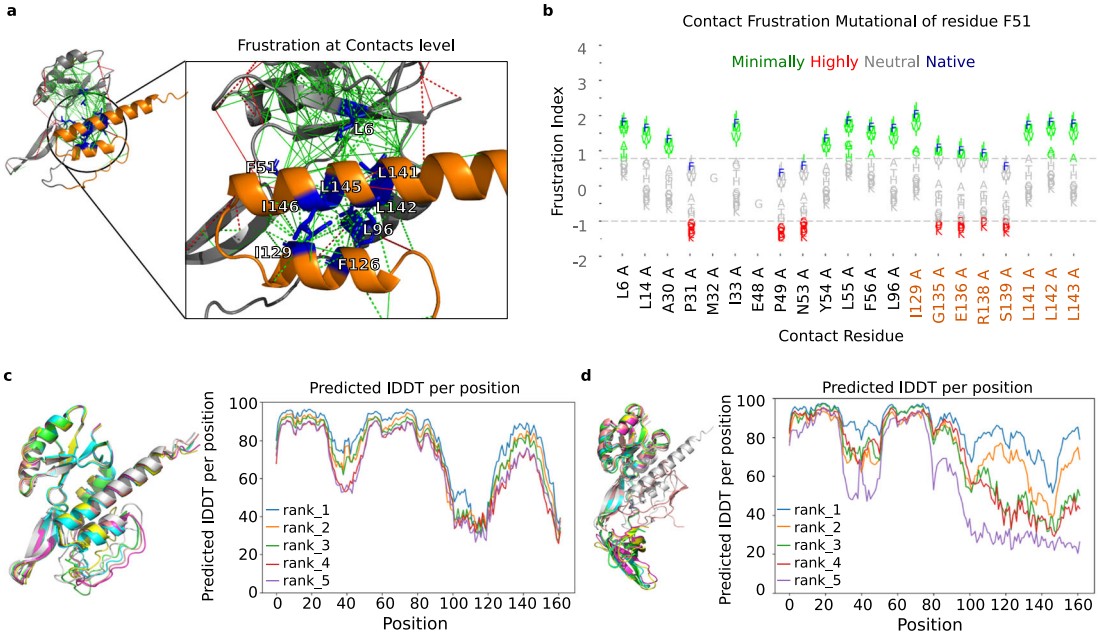

**Fig. 5 | Frustration analysis of a metamorphic protein conformational change.** **a** FrustraEvo Mutational index results. Red lines correspond to highly frustrated interactions, and green lines to minimally frustrated interactions (see Methods). Orange backbone corresponds to the interdomain region (CTD), and residues in blue and sticks are the nine interface residues. **b** Frustration changes upon mutation for Phe 51 using FrustratometeR. The *x* axis shows the residues with which the residue, either wild-type (Phe) or mutated, establishes contacts in the structure. In the *y* axis, we show the mutational frustration index for the contacts. The wild-type amino-acid identity is shown in blue, and the variants are colored according to their frustration state. **c** AlphaFold2 top five predicted models superimposed for RfaH containing different sets of mutations for SFMs and **d** HFMs. Source data are provided as a Source Data file.

(Supplementary Fig. 15a). The local frustration profiles of these mutations (Supplementary Fig. 14b) show that 4 of them are SFMs (L6V, I129V, L141V, L145I), one changes the frustration values from minimal to neutral (146F) and only one (L142S) is an HFM. When L142S alone is introduced in RfaH, AlphaFold2 returns 1 model with an αCTD fold, 2 with loop-like CTD structures and 2 with a βCTD fold (Supplementary Fig. 15a). From the 115 residues that change their identity in the RfaH/NusG sequence alignment, L142S is the only case where the αCTD changes to adopt a βCTD-like conformation (Supplementary Fig. 15b) upon AlphaFold2 structure prediction. The CTD conformation adopted by the L142S mutant is less similar to the one of NusG when compared to that obtained when the 6 NusG-like mutations are introduced. Therefore, it seems that not only is the introduction of frustration necessary to trigger a conformational change in RfaH but also there is a need to fine-tune the minimally frustrated contacts in the interdomain region.

This example illustrates how conserved frustration patterns can be used to generate hypothesis-driven experiments to modify specific biophysical properties in the context of protein engineering strategies.

## Discussion

We have introduced the analysis of energetic patterns across protein families based on the conservation of frustration levels. These patterns reveal conserved physicochemical constraints related to protein stability and function, providing a biophysical interpretation of the impact of sequence divergence over evolutionary timescales. Proteins evolve constrained by the need to minimize energetic conflicts related to folding and stability[42] while paying an energetic cost to maintain functional sites[2,43–45]. By collectively analyzing frustration in proteins with common ancestry, we have previously shown the presence of energetic constraints that exist to preserve stability and function in protein families. We observed the presence of mainly foldability constraints in the Ankyrin Repeat Protein family[24] where consensus identities in the MSAs correlated with high stability signals. As a consequence of functional promiscuity and non-conserved interaction interfaces with their targets, there are no conserved and highly frustrated signals in the family. In contrast, highly frustrated interactions are found, mainly involving the catalytic residues, in globular cases such as the Beta Lactamases[10]. But beyond catalysis, we also found noncatalytic residues being highly frustrated that when mutated would have consequences on fitness and antibiotic resistance. This suggested that frustration conservation analysis would have broader applications to the study of protein physiology.

Here, we have shown that the analysis of conserved frustration patterns in individual protein families leads to the identification of functional constraints that correlate with changes either in stability (PDZ, SH3 and KRAS, Fig. 2c, e, g), function (PDZ, SH3 and KRAS, Supplementary Fig. 2) or structural conformation (RfaH, Fig. 5). The comparison of frustration patterns across protein families points to regions of functional diversity (hemoglobins, Fig. 3, RAS family Supplementary Fig. 4 and SARS-CoV-2 Fig. 4a, b) and specificity (SARS-CoV-2, Fig. 4c). In the case of the SH3 domain, FrustIC values of minimally frustrated residues, although not perfectly, correlate with experimental ddPCA fitness values (Fig. 2c). For the metamorphic RfaH protein, the analysis of conserved frustrated contacts led to the identification of key residues involved in conformational transitions. The best example is Leu142, which is proposed to play an important role in holding the metamorphic domain from transitioning from the all-α to the all-β conformation.

In the case of the closely related α and β globin families that constitute the hemoglobin molecule, the differential roles in the transport of oxygen and contribution to the assembly of the quaternary complex is translated into a significant difference in the number and location of energetically conserved positions corresponding to protein-protein interaction sites and salt bridges

independently acquired during their divergent, evolutionary trajectories (Fig. 3). Relevant common and divergent features can be obtained even for largely diverse protein families, like the RAS superfamily, where sequence and functional conservation starts to vanish. There, conservation can be detected at the level of pairwise contacts instead of single residues, some of which constitute specific networks that are maintained in the entire RAS superfamily (Supplementary Fig. 6), while others are subfamily-specific. Some specific interactions, involving residues within the active site and associated with the common GTP-binding function, are systematically frustrated (mostly involving Asp57 and Lys117) in all families showing strong evolutionary pressure to maintain those conflicts despite sequence and functional divergence. Additionally, SDPs found in these protein families define differentially conserved specificity sites, whose interpretation in functional terms can be better understood when frustration is considered as an additional and complementary layer of information. Finally, we have shown how a comparative analysis of frustration conservation can be performed in an unsupervised and automatic manner, leading to the identification of potential functional adaptations of protein families. As an example, we performed an analysis for 22 Coronavirus protein families. Despite an observable correlation between average FrustIC and SeqIC across protein families, the relationship between these two quantities is modulated by different aspects such as the phylogenetic diversity of the family or their propensity to protein disorder. By taking these factors into account and as shown with the PLPro example, frustration conservation analysis can be used for identifying positions that are likely relevant either for stability or function.

This study does not come without limitations. For proteins with multiple conformations associated with their function, unsupervised modeling can predict distinct conformations that could be associated with different frustration patterns and therefore no energetic conservation might be observed unless the different conformations are clustered and analyzed separately, even in the presence of high sequence conservation. Future developments in our strategy should take into account the conformational diversity of the native state of proteins to account for this. In the same note, frustration states are defined according to thresholds on a continuous score (i.e. the frustration indexes). Therefore, residues with frustration values close to the thresholds that are used to define the frustration classes can show heterogeneous frustration states across proteins, while having similar continuous values. For this reason, when no frustration conservation is observed but there is a hint of functional importance, supplementary analysis on the continuous frustration scale could be useful. Also, in many of our examples, we performed unsupervised clustering of sequences to create the families datasets, as well as generated MSAs automatically. In order to exploit the capabilities of our strategy to its maximum, researchers are encouraged to invest a good amount of effort into curating the family datasets as well as manually curating MSAs so the signal is not buffered out.

The evolution of protein families is constrained by both a narrow margin of stability and foldability energetics in the context of demanding functional requirements. In these margins, too many minimally frustrated regions might hinder functional evolution, while the presence of too many highly frustrated regions will prevent folding from happening[14]. Frustration conservation analysis within protein families can be used to define the theoretical limits to preserve function throughout evolution revealing the interplay between sequence, structure(s), dynamic and function[46]. Now that high-quality protein structure models can be obtained for the members of any protein family, our frustration conservation analysis strategy stands as a valuable tool to increase the level of functional annotation in biological databases[47]. Future work would involve performing a large-scale study across protein families in databases such as CATH[48] that would permit to provide functional predicted annotations to resources like the

PDBe-KB database[49]. Similar to what was done some years ago with the analysis of frustration across large datasets of single nucleotide variants (SNVs) in humans[50], our family-derived constraints could be used to complement such analysis and provide an evolutionary context to the impact of sequence variants in different diseases. The importance of the evolutionary history of proteins in different diseases has been recently highlighted in a wide study in primates[51]. Finally, the examples shown here illustrate how the approach we present can guide applications such as the construction of artificial proteins or the prediction of the risk of emergent natural pathogenic protein variants.

## Methods

### Local frustration

The FrustratometeR algorithm allows one to localize and quantify local energetic frustration in protein structures by calculating 3 different frustration indexes, FIs[8,16]. Here we will give a summarized explanation of the methodology and a very detailed description of the method can be found in[8,52].

Given a protein structure, a set of interacting residue pairs is defined based on distance thresholds. Interactions are classified into short-range (distances between Cβ below 6.5 Å), long-range (between 6.5 and 9.5 Å) or water-mediated (long range and exposure to solvent) according to the contact distance and the solvent accessibility. This classification is important as it will define which of the coarse-grained AWSEM-MD energy functions (matrices)[53,54] will be used to measure the energy of the native interaction as well as the one of the decoys to calculate the frustration indexes.

Depending on the heuristic that is used to derive the decoys, against which the native energy will be compared to, we can obtain one of the 3 different FIs, i.e., one of the pairwise indexes (mutational or configurational) or the single residue frustration index (SRFI). For all the FIs, 2000 decoys are generated. The native energy is compared against the mean and standard deviation of such distribution by computing a Z-score and therefore, the FIs are expressed in standard deviation units, typically in the [−4, 4] range.

For the pairwise indexes, for a given pair of contacting residues, their interaction energy is compared to the energies that would be found by placing different amino acids in the same native location (mutational frustration index, the native amino acid identities are modified while the contacting residue, as well as the residues solvent accessibility values, are fixed) or by creating a different environment for the interacting pair (configurational frustration index, the native amino acid identities as well as the interacting distance and solvent accessibility values are modified). When comparing the native energy to the energy distribution resulting from these decoys, the native contacts are classified as highly, neutrally, or minimally frustrated according to how distant the native energy is from the mean value of the energy distribution of the decoys, taking into account the standard deviation of the distribution as a normalizing factor (see thresholds in the next paragraph). An analogous approach can be used to calculate the FI for single residues (single residue frustration index, SRFI). In this case, the set of decoys is constructed by shuffling the identity of only one residue, keeping all other parameters and neighboring residues in the native location, and evaluating the total energy change upon mutation, i.e., integrating the interactions that the residue establishes with all its neighbors. Specific details on how the decoys are generated are found in ref. 8.

The configurational and mutational pairwise contacts FIs have the following thresholds to define the different frustration states for the interactions, as proposed by Ferreiro et al.[8,16]: if FI < −1 then the interaction is highly frustrated. If FI > 0.78 then the interaction is minimally frustrated. If −1 < FI < 0.78 then the interaction is neutral. In the case of the SRFI thresholds for single residues, if SRFI < −1 then the interaction is highly frustrated. If SRFI > 0.55 then the interaction is minimally frustrated. If −1 < SRFI < 0.55, then the interaction is neutral.

An advantage of the simple definition of the FrustratometeR energy function is that it only has parametrization for the 20 canonical amino acids within the AWSEM-MD energy function[54]. Substrates, small ligands or any other modification to the canonical amino acids are not considered, and therefore highly frustrated signals can appear at the sites where these molecules bind or where post-translational modifications occur. An example of the latter is enzymatic sites[10]. A similar situation happens with protein-protein interactions when frustration is calculated on isolated proteins. Interactions sites, which otherwise would be energetically minimized at the quaternary structures, appear highly frustrated when frustration is calculated on the monomers alone.

### FrustraEvo's pipeline

FrustraEvo calculates how conserved local frustration is in a set of protein structures that belong to the same protein family. It can be used in two different modes depending on which FI is used: (1) Single residue mode: using the SRFI, (2) Contacts mode, which can be used with either the configurational or mutational FIs.

The input consists of (1) a MSA in FASTA format with sequences composed solely by the standard 20 amino acids code (other characters accepted by FASTA are replaced by a gap), and (2) a set of protein structures (experimental or models) in PDB format corresponding to the same set of sequences contained in the MSA.

The ID of each protein sequence within the MSA should match its corresponding structure file name (without the pdb extension). Sequences within the MSA should match exactly the ones contained in the PDB files.

Sequence and Frustration information content, SeqIC and FrustIC, respectively, are calculated using information theory concepts. The information content (IC) based on the distribution of states is calculated, as shown in Eq. (1), as the maximum possible entropy ($H_{max}$) minus the observed entropy ($H_{observed}$). SeqIC is calculated from aligned residues in the MSA, based on the distribution of amino acid identities. It is calculated by the ggseqlogo R package[55] using the MSA as input.

FrustIC is calculated for aligned residues in a MSA or equivalent contacts across proteins in the MSA, based on the frustration states mapped into the residues from the structures. A reference protein is selected to define over which residues or contacts the conservation calculations are calculated. The reference structure can be defined by the user or otherwise FrustraEvo selects the protein that maximizes the sequence coverage of the MSA. All columns in which the reference protein has a gap are removed from the MSA (ungapped MSA).

In FrustraEvo's single residue mode, the Frustration Information Content (FrustIC) for each column in the MSA is calculated based on the Shannon entropy formula:

$$H_{observed} = - \sum_{i=1}^{M} P_i \log_2 P_i \qquad (1)$$

where $P_i$ is the probability that the system is in frustrated state $i$. The probabilities are normalized such as $\sum_{i=1}^{M} P_i = 1$, where M is the number of possible frustration states. For the frustration index, we consider minimally, neutral or highly frustrated states, therefore, $M = 3$. To take into account background probabilities, the information content is calculated as:

$$FrustIC = H_{\_max} - H_{\_observed} \qquad (2)$$

Generally, it is considered that $H_{max}$ is reached for a uniform distribution of states: then $P_i^{max} = \frac{1}{M}$ and $H_{max} = log_2(M)$. Nevertheless, if states are not equally likely to occur a background probability distribution of states should be used to estimate $H_{max}$. We used the distribution of states reported by Ferreiro et al.[8] as background frequencies to calculate the $H_{max}$ for FrustIC calculations (background

frequencies as follows: minimally frustrated = 0.4, highly frustrated = 0.1, and neutral = 0.5).

Similarly to the single residue mode, in FrustraEvo's contacts mode, a reference structure is taken to define which residues will be considered to evaluate contacts across proteins. Taking in consideration that the MSA was ungapped according to the reference structure, FrustraEvo calculates the frequency of having contact between columns $i, j$ in the MSA, on each structure in the dataset, where $i, j \in [1, N]$, with $N$ being the number of columns in the ungapped MSA. As a result FrustraEvo will calculate, for each possible contact, according to pairs of columns within the ungapped MSA, the information content contributions from each frustration state. The FrustIC of a given contact will be calculated as the sum of the individual contributions from each frustration state. The background frequencies are the same as for the single residue mode. Plots for the frustration and sequence logos and contact maps are made using the ggplot2 R package.

### Data visualization
We have developed the Multiple Sequence Frustration Alignment (MSFA) visualizations to compare FrustratometeR results across multiple protein sequences (e.g., Figure 1a). This type of plot consists of a heatmap for which each cell contains the residues in the MSA. Each cell is colored according to its SRFI in the corresponding structures, i.e., minimally frustrated residues are colored in shades of green, neutral in gray, and highly frustrated in red.

In addition to the previous, we designed the Consensus MSFA to summarize and visually compare the FrustIC and SeqIC, from multiple MSFAs from multiple protein families. In this case, each cell depicts the consensus sequence of the MSA of each protein family and the letter size is proportional to its SeqIC value. The cells' background color corresponds, in shades from green through gray to red, to the median frustration value of that residue across all structures in the family. The background of the cell is white when FrustIC ≤ 0.5.

### AlphaFold2 structure predictions
Unless mentioned, for all analyses, the 3D structure of each sequence contained in the final MSAs was determined using AlphaFold2 with default parameters. The best-ranked model (rank1) was considered.

### Structural and interface mapping
All mappings between sequence data and the corresponding PDB structures and interfaces (when applicable) or Alphafold2 models were performed using the 3Dmapper tool[56].

### PDZ, SH3, and KRAS workflow/data
To analyze GRB2-SH3 and PSD95-PDZ3 protein domains and KRAS with FrustraEvo, we generated MSA alignments for each family. To this extent, the sequences of reference, chain A from PDB 2VWF, in the case of GRB2-SH3, chain A from PDB 1BE9 in the case of PSD95-PDZ3 and P01116-2 for KRAS, were blasted against the non-redundant NCBI protein database (from April 2021) using Blast v2.11.0 with default parameters. Hits with e-value ≥ 0.05, query coverage <70%, and hits containing the words "artificial", "fragment", "low quality", "partial", "synthetic" were filtered out. The remaining hits were clustered using CD-Hit[57] with default parameters. Representative sequences were then aligned using MAFFT v7.453[58] in the case of the SH3 and PDZ families and using HMMer's hmmalign method in the case of KRAS, both with default parameters. The resulting MSAs contained 173, 679, and 2500 sequences for GRB2-SH3, PSD95-PDZ3, and KRAS, respectively. For each of the sequences, AlphaFold2 models were produced. Their mean pLDDT values considering only residues that are contained in the alignment region of the reference structure were 89.3, 78.3, and 81.6 for SH3, PDZ3, and KRAS, respectively. Finally, the MSAs and the AlphaFold2 models were analyzed with FrustraEvo.

To calculate the correlation between experimentally determined stability and binding scores and FrustraEvo results, we used the ddPCA fitness scores from Faure et al.[20]. From the supplementary files provided, we used

- "JD_PDZ_NM2_bindingPCA_dimsum128_filtered_fitness_replicates. RData" and "JD_PDZ_NM2_stabilityPCA_dimsum128_filtered_fitness_replicates.RData"

to extract the results for PSD95-PDZ3 and

- "JD_GRB2_epPCA_bindingPCA_dimsum128_fitness_replicates. RData" and
- "JD_GRB2_NM2_stabilityPCA_dimsum128_fitness_replicates. RData"

to extract the results for GRB2-SH3. In all cases, when loading the data files in R, only the dataframe "singles" was used in our analysis. We modified the numbering of PSD95-PDZ3 dataframes (column "Pos") to start in 15 and end in 98 instead of 1 and 84 to match the real position of the domain in the protein sequence of reference (chain A from PDB [1BE9]). For each position of both protein domains, the mean fitness value for each protein position is considered in the comparison to SeqIC and FrustIC. For the KRAS example, fitness values were obtained from Supplementary Table 4 (sheet 2) from the work of Weng et al.[21] and the mean fitness value was calculated for position and type of assay (AbundancePCA and all BindingPCA assays). Finally, we calculated the Pearson correlation between the mean fitness stability and binding data of GRB2-SH3, PSD95-PDZ3, and KRAS individually and their corresponding FrustIC or SeqIC values computed with FrustraEvo.

### Hemoglobins workflow/data
We have retrieved all non-redundant mammalian hemoglobins ($n = 21$) present in PDB (by April 2022; Supplemental Table S6), splitted them into two non-redundant structure sets of α- and β-globins and calculated their frustration patterns using FrustratometeR[17]. Three MSAs were built, containing: all the α and β-globins together, only α-globins, and only β-globins, the last two subsetted from the first one. Finally, for each MSA, we computed SeqIC and FrustIC values using FrustraEvo single residue mode.

### RAS superfamily workflow/data
A total of 160 human protein sequence IDs of the RAS superfamily were extracted based on Fig. 3 from the work of Rojas et al.[22] Nine proteins were not included in the analysis as they were no longer present in Uniprot or their IDs did not match any entry. The sequences are grouped into ARF ($n = 26$), RAB ($n = 64$), RAN ($n = 1$), RAS ($n = 38$) and RHO ($n = 22$). Since the RAN family only contains one human sequence, it was discarded from our analysis. Sequences were aligned with MAFTT v7.453[58] with default parameters, and their models were generated with AlphaFold2 (mean pLLDT scores are ARF = 86.3, RAB = 77.4. RAS = 83.5, RHO = 82.6).

### Unsupervised analysis of the SARS-CoV-2 and related coronaviruses proteins
We retrieved all homologs to proteins within the SARS-CoV-2 proteome with known sequences across coronaviruses (see following subsections) and generated models with AlphaFold2. After filtering out those proteins for which the structural models did not have enough quality, we processed a total of 22 protein families (Supplemental Table S4). We applied the S3Det software[5] to subdivide the set of proteins into subfamilies and find their SDPs. Finally, we used FrustraEvo to obtain SeqIC and FrustIC values for each subfamily. The different steps of the pipeline are explained in more detail in the next sections.

**Coronaviruses homologs retrieval and MSAs building.** Reference sequences for all SARS-CoV-2 proteins were retrieved from the reference genome MN985325 according to the annotations of the genbank file from the supplementary data provided in the work of Gordon et al.[59] (strain USA-WA1, a file called "2020-03-04719B-Gordon et al. 2020 - SARS-CoV-2 USA-WA1 Genome Annotation.gb"), with the following correction in line 408 "13442..16236" changed for "join(13442..13468, 13468..16236)". Then each nucleotide reference sequence was translated and blasted against the non-redundant NCBI protein database (from March 2021) using Blast v2.11.0[60] with default parameters, a maximum of 100,000 hits, and e-value < 0.05, excluding taxids 2697049, 2724902, 2724903, 2724904 which correspond to SARS-CoV-2 related sequences and taxid 32630 which corresponds to artificial sequences. Afterwards, the hits that were not from the Coronaviridae family were also filtered out. Hits with less than 70% query coverage and hits containing the words "artificial", "fragment", "low quality", "partial", "synthetic", "Severe respiratory syndrome coronavirus 2" or "SARS-CoV-2" in the description were also excluded. The remaining hits were aligned using MAFFT v7.453[58] with default parameters. In the case of the non-structural proteins (nsp), some of the hits retrieved corresponded to the whole orf1ab, so the alignment was trimmed to the region containing each specific nsp according to the SARS-CoV-2 reference. Afterwards, the sequences were clustered using CD-Hit[57] with parameters -c 0.98 -s 0.90 and the representative sequences were aligned with MAFFT.

**Classification into subfamilies and SDPs detection.** The software S3Det[5] was used to classify the protein datasets into subfamilies and detect SDPs for each viral protein, following the same steps as in[61]. Supplemental Table S2 specifies the number of clusters determined per viral protein as well as the depth and length of each protein MSA.

**Structural modeling and models quality filtering.** Structure models for all proteins contained in the MSAs were generated using AlphaFold2[18] with default parameters. To ensure the high quality of the models, each MSA was trimmed to fit the length of a PDB of reference of SARS-CoV-2 proteins. When more than one PDB structure was available, the one that maximized the MSA coverage was selected. The reference PDBs for each MSA and the trimming positions are in Supplemental Table S3. Transmembrane proteins M, nsp4, and nsp6 did not have a PDB available and were removed from the analysis. Finally, only high-quality models, i.e.: mean pLDDT per SDP subfamily ≥80, were considered in the frustration analysis (Supplemental Table S4 and Supplementary Fig. 7).

**Phylogenetic balance.** We calculated the phylogenetic diversity of each S3Det cluster of the 22 families of viral proteins (Supplementary Fig. 8). The phylogenetic balance is represented by the standard deviation of the subgenus classes proportions across the cluster. When the standard deviation is low, it means that the cluster is balanced in terms of phylogenetic variability, i.e.: if there are 2 or more subgenus classes present, they are in similar proportions. On the contrary, when the standard deviation is high, the cluster is unbalanced, i.e.: the cluster is mostly represented by one of the subgenus classes.

**The PLpro example.** A total of 122 non-redundant PLPro coronavirus homologous sequences were obtained and divided into four subfamilies which coincided with the following subgenera in Betacoronavirus: Sarbecovirus ($n = 31$), Nobecovirus ($n = 11$), Merbecovirus ($n = 35$) and Embecovirus ($n = 45$) (Supplemental Table S4). In addition, and by making use of the extensive amount of experimental structures that are available for SARS-CoV-2, we retrieved all deposited PLPro structures in the PDB ($n = 29$) and used them as an additional dataset that we also processed with FrustraEvo. The rationale behind this is that analyzing local frustration patterns across multiple structures

from the same protein allows us to study the energetic determinants of the protein taking into consideration the conformational diversity of its native state. Flexible regions that exist in various frustration states across different structures will not show a high conservation signal, while those that do not vary much will.

## Metamorphic protein workflow/data

**Orthologs selection.** While *Escherichia coli* RfaH (UniProtID: P0AFW0) is the only protein with available structures for both metamorphic folds, further proteins were selected as metamorphic RfaH orthologs based on the following criteria from bibliography. Four orthologs from *Salmonella typhimurium* (sequence identity: 88%), *Klebsiella pneumoniae* (80%), *Vibrio cholerae* (64%), and *Yersinia enterocolitica* (43%) have been demonstrated to be able to substitute *E. coli* RfaH function in vivo[61]. Additionally, deleterious mutation of RfaH in *Y. pestis* and *Y. pseudotuberculosis* exhibits lipopolysaccharide defects similar to *E. coli* ΔrfaH and, therefore, metamorphic behavior[62]. Therefore, all protein sequences for these RfaH orthologs were retrieved for further analysis.

Selecting metamorphic orthologs for RfaH is not a straightforward task, as experimental confirmation is lacking for the majority of RfaH-related sequences. We first retrieved all sequences from the IPR010215 entry in the InterPro database[63] and clustered them at 90% identity using CD-HIT[56], which gave us a total of 1004 sequences. As a strategy to determine which sequences are likely to be metamorphic homologs for RfaH in *E. Coli* we followed these criteria: (i) for each RfaH sequence, there must be at least one reported NusG sequence in the same organism in the Uniprot database[64], (ii) the full-length RfaH protein sequence must be predicted to fold into the autoinhibited, α-folded C-terminal domain (CTD) structure for RfaH (PDB 5OND); and (iii) the isolated CTD of RfaH must be predicted to fold into the canonical βCTD fold (PDB 2LCL, 2JVV). The CTD was considered to start from the first residue forming a secondary structure in the PDB 2LCL (residue 115, pattern KVII). We randomly selected sequences from the 1004 total set of entries in the redundancy-reduced IPR010215 set until we completed 30 proteins that fulfilled the above-mentioned criteria. In order to assure a high-quality set of potentially metamorphic homologs, we manually confirmed that each instance to be added to the analysis fulfilled the mentioned criteria.

Finally, AlphaFold2 models (mean pLDDT = 71.97 ± 3.8) were generated for each sequence in the ortholog set; min = 65.1, max = 79.4). It is worth noticing that metamorphic proteins use to contain regions with lower pLDDT (linkers and the metamorphic domain) scores due to their conformational diversity. For this reason we considered all structures without applying any further quality filter based on their mean pLDDT score.

**Interface residues identification.** To detect those residues potentially involved in the fold-switch (the 9 interdomain residues, we selected those residues that: (1) establish interdomain residue contacts between the two domains according to the contact maps obtained with FrustratometeR and further processed by FrustraEvo; (2) are located in the metamorphic region and their FrustIC ≥ 0.5 based on the contacts mode; (3) are present in more than 50% of the analyzed models; (4) have >50% of their interactions minimally frustrated; and (5) have at least 3 minimally frustrated interactions with other CTD residues. A total of 9 interdomain residues satisfied the previous criteria: L6, F51, L96, F126, I129, L141, L142, L145, I146.

**SFM and HFM *E. coli* RfaH mutant sequences.** For each interdomain residue that was detected, as mentioned before, we used the FrustratometeR module to predict the frustration change upon mutating the native identity by all the other possible non-native amino acids. For each residue, we selected an amino acid identity that would maintain a frustration value as similar as possible to the native identity (SFM) and another one that would introduce as much frustration as possible

(HFM). We generated RfaH mutants containing both the set of 9 SFMs and the 9 HFMs.

**NusG-like RfaH.** We aligned the RfaH and NusG sequences from *E. Coli* (Supplementary Fig. 13a) and observed that 6 out of 9 of the previously identified interdomain residues have different identities between the two proteins. We generated a "NusG-like" RfaH sequence by replacing the 6 residue identities from NusG into RfaH.

### Computational infrastructure and software requirements
Data was processed using the Marenostrum4, Minotauro, and Power9 supercomputers at the Barcelona Supercomputing Center. Computational resources from CCAD-UNC, which is part of SNCAD-MinCyT, Argentina were also used. Plots were produced with ggplot2 and ggpubr R packages. In this project, we have used the FrustratometeR package to calculate the local frustration patterns for all the presented analysis[17].

### Reporting summary
Further information on research design is available in the Nature Portfolio Reporting Summary linked to this article.

## Data availability
All input data needed to reproduce the main results of this article as well as the intermediate outputs are available at this ZENODO repository https://zenodo.org/records/10093060, https://doi.org/10.5281/zenodo.10093060). Source data are provided in this article for figs. 2A–G, 3A–C, 4A, B, 5A–C, 6A, B Source data are provided with this paper.

## Code availability
FrustraEvo code, written in Python 3 and R 4.1.2 programming languages, is available at: https://github.com/proteinphysiologylab/FrustraEvo. A Docker container is also available at (https://hub.docker.com/r/proteinphysiologylab/frustraevo).

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

## Acknowledgements

RGP thanks Professor Christine Orengo and Nicola Bordin for early discussions and data provision. RGP holds a fellowship from Grant IHMC22/00007 funded by the Instituto de Salud Carlos III (ISCIII) and by the European Union NextGenerationEU/PRTR. C.P. is supported by the fellowship "Juan de La Cierva - Formación" from the Spanish Ministry of Education and Science (ref. FJC2021-046655-I). DUF and MIF are supported by the Consejo de Investigaciones Cientificas y Tecnicas (CONICET). PGW is supported by the Center for Theoretical Biological Physics sponsored by the NSF grant PHY-2019745 and the D.R. Bullard-Welch Chair at the Rice University Grant C-0016.

## Author contributions

M.I.F and V.R. contributed equally. R.G.P conceived the project. R.G.P and A.V. supervised research. M.I.F. and V.R. developed software and provided code to produce results. M.I.F, V.R-S, C.P, M.R-D, R.G.P, D.U.F, P.W. and A.V. contributed to the analysis and interpretation of data of all sections. C.R.-S. and P.G-D mainly contributed data and discussed the analysis of the Metamorphic proteins section. M.M. and C.D.S. mainly contributed data and discussed analysis to the hemoglobins section. R.G.P., V.R and M.I.F. wrote the original draft. All authors participated in substantial discussions and edited and approved the final version of the manuscript.

## Competing interests

The authors declare no competing interests.
