## [Peer Review File · Nature Communications]

Local Energetic Frustration Conservation in Protein Families and SuperfamiliesREVIEWER COMMENTS

Reviewer #1 (Remarks to the Author):

This review offers innovative insight into protein families, their functional diversity, and conservation. The article presents an original approach to analyzing energetic patterns across protein families based on the conservation of frustration levels, providing new insights into protein stability, function, and evolution. It is dense and technical, so simplification and reduction in the Results section would be beneficial. Additionally, the authors could consider putting their work in a broader context by comparing their findings with the existing structural biology literature on the proteins they discuss, making the work more accessible to biologists.

Understanding the concept of frustration in proteins is crucial. Highly frustrated proteins are unstable, which could lead to conformational changes related to allosteric regulation. In contrast, minimally frustrated proteins have lower energy and are stable. It's also essential to understand the key metrics introduced here— FrustrIC and SeqIC (the conventional approach), and their differences. FrustrIC measures the conservation of frustration levels in protein families. A high FrustrIC score indicates that the frustration level of a particular residue or interaction is highly conserved throughout the family. SeqIC measures the conservation of sequence identity. The authors provide well-documented examples to demonstrate their approach's utility, including α and β globin, the RAS, and Coronavirus protein families. Potential applications range from improving functional annotation to engineering.

The manuscript could benefit from further discussion on the potential limitations of the frustration conservation analysis and how these limitations may affect the interpretation of the results. Figure 2 shows that the correlation between abundance fitness (measured by ddPCA) and FrustrIC is good but not superb. Regarding the interpretation of Figure 2, if a protein is unstable, it is more susceptible to degradation, and hence has a lower abundance.

One key takeaway from this work is that quantifying the frustration pattern in a protein family with common ancestry can reveal information about the evolutionary importance of certain sequences. A conserved interaction with a low frustration index indicates that it is energetically favorable and may play a crucial role in the stability or function of proteins across the family. Conversely, a conserved interaction with a high frustration index suggests that despite being unfavorable, this interaction is maintained in the family members possibly for maintaining a specific function. The manuscript could be even clearer on this point.

While the authors identified important residues, they could also discuss their findings in the context of existing literature. There are plenty of articles on the activation mechanism of RAS that the authors could

compare their work with, linking it to function. As to the last subsection of the Results section on engineering conformational changes, the text could benefit from simplification and further discussion on their biological significance. As we noted above, along these lines, the article might also provide a clearer comparison of the frustration conservation analysis with other existing methods, highlighting the advantages and potential limitations of the approach.

I noticed an obvious error. On page 8, Glu carries negative charges and is obviously not neutral, as stated in the text. According to the sequence of Hemoglobin subunit beta, instead of Glu39, it should be Gln40. See <https://www.uniprot.org/uniprot/P68871/entry#sequences>

Overall, these points may improve the manuscript and help a reader to better understand the innovative implications of this work.

Reviewer #2 (Remarks to the Author):

Overview of paper:

This paper considers the use of the concept of ‘energetic frustration’ and its quantification, in an analysis of protein sequence evolution, giving 3 well worked families as exemplars and applying it to the proteins of the SARs-COV2 virus. They also use it to try to predict mutations which will result in morphological changes in a protein known to change fold as part of its function. Their method utilises predicted protein structures from AlphaFold to generate ‘frustration indices’ and then sees how they change/are conserved during evolution. The authors have also generated a basic web site to allow anyone to apply their method to any protein sequence.

The concept of frustration was first developed by Wolynes and colleagues in 2007. This paper, involving groups in Argentina, Spain and Chile, (including several authors from the Wolynes group) seeks to employ this concept and use three ‘frustration parameters’ derived from predicted 3D structure, to quantify frustration, as an additional ‘feature’ to analyse protein family evolution and MSAs. Alongside the usual treatment of protein sequence evolution using MSAs, they use predicted 3D structures (from AlphaFold) to calculate the frustration indices and then use them to cluster sequences, equivalent to the way that conservation scores and co-variance are used to help to predict protein function.

My comments are below:

- The paper seems to focus on the development of a new method for variation analysis, rather than asking the fundamental question – is ‘frustration’ conserved during evolution and if so, why? Please address this point in more detail for each of the 3 families described, which appear to have very different characteristics. Surprisingly they do not directly address the challenge of using frustration to predict the impacts of variation.
- More information is needed in paper on how the frustration index is calculated – this is the basis for the whole analysis. I realise there have been several papers on this, but for such a general paper, it would really help to provide more detail in the methods, rather than just referring to other papers.
- There are MANY methods for variation analysis available. How does this approach compare with simple MSA sequence analysis? This is implicit in some of the graphs but could be addressed head-on. It would be useful to provide some numbers about for example the ability to identify pathogenic variants. A general conclusion here would help to transcend the details to reveal some fundamental principles.
- They have defined three indices(FIs) of frustration – mutational, configurational and single residue (SRFI) from the predicted protein structure. They also measure the information content of a MSA ‘column’ – calculating FrustrIC and SeqIC (in an R package) – the sequence information content. This has all been previously published. The new work is the analysis of ‘evolution’ of frustration.
- The analysis of the three families is helpful – and they illustrate that sometimes frustration is conserved, sometimes not.
- I am somewhat confused by the impact of the ligands on ‘frustration’. Clearly the Lys117 in KRAS is involved in binding the nucleotide through electrostatic interactions. Is frustration calculated with or without the ligand being present? Presumably the interactions are specific and required for binding. This is barely mentioned and surely is an important component to explain the results. Simple sequence analysis, not taking into account the ligands, can be misleading. Of course, AF structures do not contain ligands, but functionally they are critical.
- SH3 & PDZ domains – these are protein binding domains, which are used often in biology, but individual proteins are often specific for a given PPI. Since the ligand varies across these families, then we would not expect the binding parts to be conserved – indeed they will vary exactly to produce the required specificity. This is barely discussed for these families and should be. This can be described as ‘family promiscuity’, rather than promiscuity of an individual sequence to bind different proteins. I do not know the details, but this aspect is relevant to conservation and should be discussed in looking at correlations. The KRAS domain is a GTPase which binds a nucleotide substrate. There should be a paragraph introducing the functions of all three domains – explaining their individual and family promiscuity and binding sites, so that the results can be more easily understood.
- The authors consider the correlation between FrustrIC (and SeqIC) and experimental stability (measured by ddPCA) for their example domain families, for which experimental data on mutagenesis and stability are available. There is correlation between stability and FrustrIC for SH3 and PDX domains: this is slightly better than that found just using sequence information (SeqIC). That said, the distributions

for the SeqIC look more convincing. For the third family, KRAS, the correlations are poor for both SeqIC and for FrustrIC, although FrustrIC highlights one residue involved in substrate binding.

- The authors then consider how frustration patterns reveal lineage specific function adaptations. For this they first consider the alpha and beta globin subfamilies. These globins form obligate tetramers (2 α +2 β). They find 12 highly frustrated position in the alpha chain; 8 in the beta chain, most of which do not overlap. Some correspond to changes in the sequence. The interaction interfaces in the tetramer are not identical in the alpha and beta chains, so does this explain fully the differences? Is the Bohr effect over and above this – and is the critical residue identified just by sequence analysis without consideration of frustration? They then move on to analysing members of the 4 RAS sub families. The SRFI is not conserved between families, even in the 5 binding motifs. When treated separately the subfamilies reveal some highly frustrated sites – mainly connected to binding to the ligand. It is not clear to me exactly what the frustration analysis reveals over and above the SDPs and the identification of the binding site. Again metal-ion binding is highlighted – is this because the energetics do not include consideration of the ligands?
- The authors then look at the proteins of the SARS-COV2 virus from the perspective of frustration. They then included analysis of the Sarbecovirus and the PLpro domain of viral polyproteins, giving a detailed but inconclusive description of their observations.
- Lastly they have performed some ‘theoretical’ mutagenesis experiments, identifying ‘frustrated’ residues and mutating them to induce a change of ‘fold’ in RfaH. They tested their prediction by running AlphaFold on the mutated sequences and inspecting the structures predicted.

In summary, this is a dense detailed paper, studying many families and subfamilies, from which it is quite challenging to draw conclusions, outside of specific observations for the specific families. I have read it several times and still struggle with the details and what insights can be drawn from using these frustration indices, that are not apparent from just using the sequences alone. The overall picture is still quite confusing. Perhaps some of the examples and detailed discussions of specific residues could be removed to give more clarity?

This approach is another way to consider the problem of variation, that builds on recent advances in protein 3D structure prediction and how we can use structural data to provide additional insight. It is still challenging to identify accurately which variations are pathogenic – and I was surprised, given the scope of this paper, that the authors did not include this in their considerations. The results are complicated, vary between families, and are difficult to both understand and consider more broadly. I think the authors included too many examples, each of which requires biological understanding, and in so doing make the paper very difficult to read and comprehend. My main question remains ‘Is frustration conserved’ – one would have thought that unless absolutely needed, then the protein would evolve itself to a more ‘comfortable’ energetically favourable state. A clear statement on what the authors have found from their analyses would help.

Reviewer #3 (Remarks to the Author):

In this manuscript, the authors integrate previously developed frustration calculation methods with new frustration conservation parameters to study the conservation of frustration in protein families and discuss the connection of such frustration with protein function. This is a new and interesting approach.

Comments:

(1) In order to make the manuscript broadly accessible (beyond bioinformaticians), the results section should be expanded and all terms need to not just be defined (in some places that has been omitted) but also explained.

(2) Similar to the above, frustration is still not commonly understood or used in the context of proteins or protein families. A few paragraphs explaining the term, how it is calculated and why frustration may occur in functional regions should be given with extensive references.

(3) The authors focus on stability and function but mention foldability only in passing. While in other protein families, foldability is a constraint, in fold-switchers, (and potentially other proteins that undergo conformational transitions), foldability is likely selected for. In this context, it would be useful to understand, what signals of foldability are likely to be seen by the frustration calculation methods used in the manuscript and what are likely missed. A larger discussion of foldability and frustration should also be given.

(4) Methods complementary to frustration analysis such as SCA and DCA (these are just examples) have been used to study protein families. These should be discussed, cited and contrasted with the frustration analysis.

Barcelona, Spain, 07/09/2023

After assessing the editorial and reviewers' comments we are very pleased to submit the response to reviewers as well as our revised manuscript. Based on the comments of one of the reviewers we have changed the title to "Local Energetic Frustration Conservation in Protein Families and Superfamilies" as we think communicate the concepts that we are reporting in a more accurate way.

We deeply appreciate the reviewers for the constructive comments that led us to significantly improve our article.

We address here the reviewer comments and we hope we have been able to meet their requests. We set the color of our responses to blue to facilitate their reading. Changes in the main text are coloured in red.

REVIEWER COMMENTS

Reviewer #1 (Remarks to the Author):

This review offers innovative insight into protein families, their functional diversity, and conservation. The article presents an original approach to analyzing energetic patterns across protein families based on the conservation of frustration levels, providing new insights into protein stability, function, and evolution. It is dense and technical, so simplification and reduction in the Results section would be beneficial. Additionally, the authors could consider putting their work in a broader context by comparing their findings with the existing structural biology literature on the proteins they discuss, making the work more accessible to biologists.

Understanding the concept of frustration in proteins is crucial. Highly frustrated proteins are unstable, which could lead to conformational changes related to allosteric regulation. In contrast, minimally frustrated proteins have lower energy and are stable. It's also essential to understand the key metrics introduced here— FrustrIC and SeqIC (the conventional approach), and their differences. FrustrIC measures the conservation of frustration levels in protein families. A high FrustrIC score indicates that the frustration level of a particular residue or interaction is highly conserved throughout the family. SeqIC measures the conservation of sequence identity. The authors provide well-documented examples to demonstrate their approach's utility, including α and β globin, the RAS, and Coronavirus protein families. Potential applications range from improving functional annotation to engineering.

The manuscript could benefit from further discussion on the potential limitations of the frustration conservation analysis and how these limitations may affect the interpretation of the results. Figure 2 shows that the correlation between abundance fitness (measured by ddPCA) and FrustrIC is good but not superb. Regarding the interpretation of Figure 2, if a protein is unstable, it is more susceptible to degradation, and hence has a lower abundance.

We thank the reviewer for pointing at this. The correlation that we obtained between the ddPCA abundance score and FrustrIC was surprisingly high (although not superb, true) to us given that the ddPCA represents a global stability measure in a single protein while FrustrIC evaluates the conservation of a local measure across homologous proteins in different species. Certainly, there might be other factors that account for the changes in ddPCA upon mutation that are not captured by our analysis. We have added the following text to reflect on this in the results section for this analysis (section 2, page 5-6).

“The correlation values between ddPCA scores and FrustrIC are not that high as they could be (~0.8 for PDZ and SH3 and 0.47 for KRAS) meaning that other factors beyond local frustration need to be taken into consideration to predict global stability as captured by ddPCA. However, the correlation is good enough to showcase the usefulness of FrustrIC as a decent in silico prediction of stability related to specific residues when no experimental data is available.”

We have also discussed the limitations of our methodology at the discussion section:

“This study does not come without limitations. For proteins with multiple conformations associated with their function, unsupervised modeling can predict distinct conformations that could be associated with different frustration patterns and therefore no energetic conservation might be observed unless the different conformations are clustered and analyzed separately, even in the presence of high sequence conservation. Future developments in our strategy should take into account the conformational diversity of the native state of proteins to account for this. In the same note, frustration states are defined according to thresholds on a continuous score (i.e. the frustration indexes). Therefore, residues with frustration values close to the thresholds that are used to define the frustration classes can show heterogeneous frustration states across proteins, while having similar continuous values. For this reason, when no frustration conservation is observed but there is a hint of functional importance, supplementary analysis on the continuous frustration scale could be useful. Also, in many of our examples, we performed unsupervised clustering of sequences to create the families datasets, as well as generated MSAs automatically. In order to exploit the capabilities of our strategy to its maximum, researchers are encouraged to invest a good amount of effort into curating the family datasets as well as to manually curate MSAs so the signal is not buffered out”

“

One key takeaway from this work is that quantifying the frustration pattern in a protein family with common ancestry can reveal information about the evolutionary importance of certain sequences. A conserved interaction with a low frustration index indicates that it is energetically favorable and may play a crucial role in the stability or function of proteins across the family. Conversely, a conserved interaction with a high frustration index suggests that despite being unfavorable, this interaction is maintained in the family members possibly for maintaining a specific function. The manuscript could be even clearer on this point.

We thank the reviewer for this observation and we agree with this. Following the reviewer's suggestion we have included the following text in the introduction to make these statements more explicit and clear to the readers:

“Our rationale is that the conservation of minimally frustrated interactions within a protein family over extended evolutionary timescales implies their crucial role in foldability or local stability. Furthermore, the conservation of highly frustrated interactions suggests that such local, unfavorable energetics conditions are required by specific functional requirements that have persisted over the evolutionary history of the family.”

While the authors identified important residues, they could also discuss their findings in the context of existing literature. There are plenty of articles on the activation mechanism of RAS that the authors could compare their work with, linking it to function. As to the last subsection of the Results section on engineering conformational changes, the text could benefit from simplification and further discussion on their biological significance. As we noted above, along these lines, the article might also provide a clearer comparison of the frustration conservation analysis with other existing methods, highlighting the advantages and potential limitations of the approach.

We thank the reviewer for their comment. Following the reviewer’s suggestion we have summarized the RAS section in the main text and moved a substantial part of it to a supplementary note where we have expanded the explanations and interpretation of results hoping the reviewer will find it more complete now.

Related to the comparison with other methods. We have included a comparison with SDPs analysis for the comparison between alpha and beta globins where we explain the advantages and limitations of frustration compared to this sequence based method and how these two are in fact complementary techniques (section 3, page 7-8).

“To further study such types of positions we have used the S3Det software to detect SDPs between the two globin lineages and to analyze their relationship with frustration conservation patterns. There are 14 SDPs between α and β globins from which only 6 differ in their frustration states between the two lineages (Table S1). Interestingly, such energetics is not trivially explainable from sequence identity. Some SDPs maintain consistent energetic levels and conservation despite changing identity. For example, position 32 shows minimal frustration both in α globins (F) and in β globins (L). Other SDPs differentiate clusters based on frustration levels, like position 57 being neutrally conserved (S) in α globins and maximally frustrated (N) in β globins. However, some SDPs of the same amino acid type, expected to have similar energetics, do not exhibit consistent frustration conservation, such as position 140 being minimally frustrated (V) in α globins and neutrally frustrated (A) in β globins. However, frustration conservation analysis has limitations. Different conformations of the same protein can yield varying frustration values, indicating the capture of distinct conformational states across families and leading to low FrustrIC values. Combining sequence variability analysis, like SDP, with frustration conservation analysis provides insights into the stability and functional consequences of evolutionary divergence between protein families. “

Related to RfaH, we have included some extra biological interpretation of the results (section 5, page 11-12). Given that we have simplified the RAS section, we have not substantially simplified the technical part of the RfaH section as we believe it is quite innovative to be used in other systems by researchers who read this work.

I noticed an obvious error. On page 8, Glu carries negative charges and is obviously not neutral, as stated in the text. According to the sequence of Hemoglobin subunit beta, instead of Glu39, it should be Gln40. See <https://www.uniprot.org/uniprot/P68871/entry#sequences>

We thank the reviewer for noticing this error and we apologize for not seeing it before. It seems that we made a mistake when converting from the one letter to the three letters code. As the reviewer says, the correct amino acid is Gln. However, while the position in the Uniprot entry is 40, at the PDB the position is 39 as that is the numbering in the PDB that we used as a reference (Human hemoglobin, PDB ID: 2dn1). We have now corrected the text and the corresponding figure legend, mentioning that we used the PDB numbering, as it is used to reference positions in the structural part. Just in case, when we mentioned the word neutral we referred to the frustration state that is conserved and not the neutral charge of the residue.

Overall, these points may improve the manuscript and help a reader to better understand the innovative implications of this work.

We thank the reviewer again as we consider that the manuscript has been improved thanks to these suggestions and comments hoping the reviewer agrees with this new version.

Reviewer #2 (Remarks to the Author):

Overview of paper:

This paper considers the use of the concept of 'energetic frustration' and its quantification, in an analysis of protein sequence evolution, giving 3 well worked families as examples and applying it to the proteins of the SARs-COV2 virus. They also use it to try to predict mutations which will result in morphological changes in a protein known to change fold as part of its function. Their method utilises predicted protein structures from AlphaFold to generate 'frustration indices' and then sees how they change/are conserved during evolution. The authors have also generated a basic web site to allow anyone to apply their method to any protein sequence.

We would like to make a small clarification here. Our method relies on either experimental structures (the globins case for example is only based on experimental structures) or models. Thanks to the reviewer comment we have now slightly modified the introduction to make it more clear. We have moved one sentence to the very end to highlight why our method will really benefit the community by exploiting the latest developments in the field of protein structure predictions.

The concept of frustration was first developed by Wolynes and colleagues in 2007. This paper, involving groups in Argentina, Spain and Chile, (including several authors from the Wolynes group) seeks to employ this concept and use three 'frustration parameters' derived from predicted 3D structure, to quantify frustration, as an additional 'feature' to analyse protein family evolution and MSAs. Alongside the usual treatment of protein sequence evolution using MSAs, they use predicted 3D structures (from AlphaFold) to calculate the frustration indices and then use them to cluster sequences, equivalent to the way that conservation scores and co-variance are used to help to predict protein function.

My comments are below:

- The paper seems to focus on the development of a new method for variation analysis, rather than asking the fundamental question – is ‘frustration’ conserved during evolution and if so, why? Please address this point in more detail for each of the 3 families described, which appear to have very different characteristics. Surprisingly they do not directly address the challenge of using frustration to predict the impacts of variation.

We thank the reviewer for their comment. Our work addresses the conservation of frustration levels within groups of extant and evolutionarily related proteins. The result informs on the concept of conservation being present because of constraints that have persisted during evolutionary times since these proteins diverged from their common ancestor. Such constraints can be classified as responding to stability (minimally frustrated and conserved) or function (highly frustrated and conserved) requirements. Because we are analyzing extant proteins, we cannot really answer the question of what has happened “during” evolution. That is what is described in the abstract and the paper. After this comment from the reviewer we consider that perhaps the title leads to a potential confusion and therefore we propose to change it to: "Local Energetic Frustration Conservation in Protein Families and Superfamilies"

A follow up including the study of these patterns on ancestral proteins would help to go into that direction. However, such analysis is beyond the scope of this work. We expect that things that are conserved across extant family members will be invariant “during” evolution and we hope to be able to analyze and interpret changes for those regions that are not energetically conserved. Following the reviewer’s suggestions we have included further details in the text to discuss the reasons for the frustration conservation patterns in the analyzed families.

Regarding the prediction of the impacts of variants, although the system is very well known and there is a working hypothesis to narrow down the effect of mutations, frustration alone is not able, to the best of our studies, to predict the impact of single variants. However, some years ago, there was an article that addressed the impact of sequence variants on frustration patterns by analyzing different sets of single nucleotide variants (SNVs), “Localized structural frustration for evaluating the impact of sequence variants. Kumar et al NAR 2016”. As the SNVs analyzed by the authors were classified according to the impact on the patients from which they were retrieved, such phenotype prior information allowed the authors to classify the SNVs according to their frustration patterns. But again, this is because there is a prior that allows for such analysis. Something similar we did for the metamorphic protein (RfaH) to select which residues were important to trigger the conformational change, we had a prior based on the biological knowledge of the system to guide us through our study. We are afraid that an automatic analysis to predict such impact is out of reach right now and all we can do is to report which positions or contacts are constrained or not. Future work to link the results of FrustraEvo and the assessment of evolutionary constraints derived from it and the phenotypic impact of SNVs at the genomic level would be of great interest and we have included it now in the discussion.

We have added this last part to the discussion:

“Similar to what was done some years ago with the analysis of frustration across large datasets of single nucleotide variants (SNVs) in humans, our family derived constraints could be used to complement such analysis and provide an evolutionary context to the impact of sequence variants in different diseases. The importance of the evolutionary history of proteins in different diseases has been recently highlighted in a wide study in primates .“

- More information is needed in paper on how the frustration index is calculated – this is the basis for the whole analysis. I realise there have been several papers on this, but for such a general paper, it would really help to provide more detail in the methods, rather than just referring to other papers.

We really thank the reviewer for pointing this out. It is true that frustration was explained superficially and in fact, it was explained in such a way in the introduction, results and method. The lack of details responds to us being very used to using and discussing these terms while it is true that it is not so common for most readers. We have now included more details on frustration in the methods section while taking away the shallow explanations elsewhere.

- There are MANY methods for variation analysis available. How does this approach compare with simple MSA sequence analysis? This is implicit in some of the graphs but could be addressed head-on. It would be useful to provide some numbers about for example the ability to identify pathogenic variants. A general conclusion here would help to transcend the details to reveal some fundamental principles.

We have answered in previous items about our impossibility to identify pathogenic variants. This is something that could be addressed in combination with other methods and prior knowledge about the variants. We included a small paragraph in the discussion but it is outside of the limits of this work that aims to introduce the general methodology to identify family constrained regions in protein families and divergent ones between evolutionary related ones.

Related to sequence variation analysis we have included a comparison between SDPs and our analysis in the globins example, highlighting the advantages and limitations of frustration and how it complements with sequence based strategies.

“To further study such types of positions we have used the S3Det software to detect SDPs between the two globin lineages and to analyze their relationship with frustration conservation patterns. There are 14 SDPs between α and β globins from which only 6 differ in their frustration states between the two lineages (Table S1). Interestingly, such energetics is not trivially explainable from sequence identity. Some SDPs maintain consistent energetic levels and conservation despite changing identity. For example, position 32 shows minimal frustration both in α globins (F) and in β globins (L). Other SDPs differentiate clusters based on frustration levels, like position 57 being neutrally conserved (S) in α globins and maximally frustrated (N) in β globins. However, some SDPs of the same amino acid type, expected to have similar energetics, do not exhibit consistent frustration conservation, such as position 140 being minimally frustrated (V) in α globins and neutrally frustrated (A) in β globins. However, frustration conservation analysis has limitations. Different conformations of the same protein can yield varying frustration values, indicating the capture of distinct

conformational states across families and leading to low FrustrIC values. Combining sequence variability analysis, like SDP, with frustration conservation analysis provides insights into the stability and functional consequences of evolutionary divergence between protein families.”

- They have defined three indices (FIs) of frustration – mutational, configurational and single residue (SRFI) from the predicted protein structure. They also measure the information content of a MSA ‘column’ – calculating FrustrIC and SeqIC (in an R package) – the sequence information content. This has all been previously published. The new work is the analysis of ‘evolution’ of frustration.

- The analysis of the three families is helpful – and they illustrate that sometimes frustration is conserved, sometimes not.

- I am somewhat confused by the impact of the ligands on ‘frustration’. Clearly the Lys117 in KRAS is involved in binding the nucleotide through electrostatic interactions. Is frustration calculated with or without the ligand being present? Presumably the interactions are specific and required for binding. This is barely mentioned and surely is an important component to explain the results. Simple sequence analysis, not taking into account the ligands, can be misleading. Of course, AF structures do not contain ligands, but functionally they are critical.

We thank the reviewer for making this comment that is fundamental for the readers to understand what our method does and why it is useful. The FrustratomeR algorithm is a coarse grained algorithm that only has the 20 standard amino acids being parametrized within its hamiltonian. This simplification of what a protein structure can contain is the core of its power to predict highly frustrated interactions. When a ligand is bound to a protein, it compensates the energetics of the interacting residues that otherwise would be in conflict with their local structure when the ligand is absent. This is the central idea behind protein-protein interaction sites being frustrated when monomers are processed alone. Those same regions are energetically favored when the quaternary complexes are analyzed as a whole (Ferreiro et al, PNAS 2007; Parra et al, Plos Comp Biol 2015). Similarly, we have found that catalytic sites and co-factor binding sites are highly frustrated for the same reasons, substrates and cofactors are not parametrized within the FrustratomeR hamiltonian and therefore are invisible to the algorithm that predicts an energetic, non compensated conflict where the ligand would bind (Freiberger et al 2019). We have added this to the methods section and briefly at the introduction hoping that this is clearer now.

- SH3 & PDZ domains – these are protein binding domains, which are used often in biology, but individual proteins are often specific for a given PPI. Since the ligand varies across these families, then we would not expect the binding parts to be conserved – indeed they will vary exactly to produce the required specificity. This is barely discussed for these families and should be. This can be described as ‘family promiscuity’, rather than promiscuity of an individual sequence to bind different proteins. I do not know the details, but this aspect is relevant to conservation and should be discussed in looking at correlations. The KRAS domain is a GTPase which binds a nucleotide substrate. There should be a paragraph introducing the functions of all three domains – explaining their individual and family promiscuity and binding sites, so that the results can be more easily understood.

We thank the reviewer for bringing this up. This comment helped us to put the results in the context of our previous articles where we briefly introduced some initial concepts by analyzing the Ankyrin Repeats Protein family (ANKs, Parra et al; Plos Comp Biol 2015) and two enzymatic families (Beta Lactamases and Aldolases, Freiburger et al PNAS 2019).

We have added the following text to the article to make these concepts more explicit and clear:

“Frustration signals conserved at the family level have appeared in the ancestor of the family and have been maintained invariantly since then for foldability or stability (minimally frustrated levels) or functional requirements (highly frustrated levels). Conversely, if a position shows large variability, it suggests that no strong constraints exist in that position and therefore the family is allowed to diversify. In the case of protein interactors, like the SH3 and PDZ domains, the binding interfaces have adapted to the binding of different partners or ligands by each family member, rendering a lack of conservation of highly frustrated positions. On the contrary, what remains conserved across the family are the common folding and stability properties, that are detected as minimally frustrated positions. A similar situation was observed for members of the Ankyrin repeat protein family ²¹ that mainly function as protein-protein interactors. In contrast, KRAS has a localized function to bind a nucleotide and cofactors, probably involving other mechanisms such as allosteric regulation. Because these functional signals are localized consistently in specific sets of residues within the MSA, their conflictive signals affect the overall stability of the protein, being a cause for the lower FrustrIC-ddPCA correlation. We have also reported similar trends for two enzymatic protein families, i.e. Beta Lactamases and Aldolases ¹⁰”

- The authors consider the correlation between FrustrIC (and SeqIC) and experimental stability (measured by ddPCA) for their example domain families, for which experimental data on mutagenesis and stability are available. There is correlation between stability and FrustrIC for SH3 and PDX domains: this is slightly better than that found just using sequence information (SeqIC). That said, the distributions for the SeqIC look more convincing. For the third family, KRAS, the correlations are poor for both SeqIC and for FrustrIC, although FrustrIC highlights one residue involved in substrate binding.

We thank the reviewer for this observation. We want to stress that frustration is a local measurement while ddPCA is a global one. This being said, it was very surprising to us to observe such a correlation showing a link between our calculations that can be run from start to end, in an almost unsupervised manner, in hours vs laborious experimental work. We have added the following text to that section as this issue was also raised by Reviewer 1.

“The correlation values between ddPCA scores and FrustrIC are not that high as they could be (~0.8 for PDZ and SH3 and 0.47 for KRAS) meaning that other factors beyond local frustration need to be taken into consideration to predict global stability as captured by ddPCA. However, the correlation is good enough and significant to showcase the usefulness of FrustrIC as a decent in silico prediction of stability related to specific residues when no experimental data is available. ”

- The authors then consider how frustration patterns reveal lineage specific function adaptations. For this they first consider the alpha and beta globin subfamilies. These globins form obligate tetramers (2 α +2 β). They find 12 highly frustrated positions in the alpha chain; 8 in the beta chain, most of which do not overlap. Some correspond to changes in the sequence. The interaction interfaces in the tetramer are not identical in the alpha and beta chains, so does this explain fully the differences? Is the Bohr effect over and above this – and is the critical residue identified just by sequence analysis without consideration of frustration?

Sequence analysis can tell us that certain positions are differentially conserved between families (SDPs). However, it cannot tell us which is the energetic signature of the SDP on each of the clusters. In the examples, we show that SDPs coincide many times with positions in the MSA where one of the two families has a conserved and highly frustrated position while the other does not.

This was addressed in the SDP text that we included, as a response to one of the Reviewer 1 requirements, before related to the globins.

We have added the following text to better explain our results in the context of the interfaces in the tetramer as well as the Bohr effect

“As mentioned earlier, highly frustrated interactions are usually suggestive of local, functional requirements. In total, there are 12 highly frustrated positions in α globins (mean FrustrIC=0.87, Fig. 3B) and 8 in β globins (mean FrustrIC=0.88, Fig. 3C) with only two residues (Q54 α , K59 β and Y140 α , Y145 β) being common to both families. This points out at differential functional adaptations that have happened independently within each lineage after diverging from their common ancestor. Several of these loci correspond to residues involved with the asymmetric interactions of each subunit within the tetrameric structure of Hemoglobin, i.e. K39 α , Y42 α , W37 β , N57 β , E101 β and N108 β . Other highly frustrated residues correspond to the differential function and structural details of each subunit type, e.g. K99 α and S124 α interact with the α Hb-stabilizing protein (AHSP), a chaperone that prevents α -globin toxicity when isolated 24,25 (Table S2), a function that is not shared with the β subunits. In addition, K7 α , E27 α and E30 α that form intra-subunit salt bridges (as shown in Fig. 3) 26 that are critical for allostery and the Bohr effect as explained by Perutz 27 (Fig. 3B, Table S2). The Bohr effect is tightly related to allosteric tensed (T) or relaxed (R) states equilibria, which shifts towards the T state due to reduced pH and higher CO₂ partial pressure, resulting in better oxygen release in the tissues. Therefore it is tightly related to the α/β interface. Indeed, several of the highly frustrated with high FrustrIC values residues identified in both subunits are key for the switch such as K40 α , Y42 α and W37 β and E101 β . Interestingly, other residues showing high SRFI and FrustrIC, such as K127 α or N57 β are at positions where mutants have been shown to subtly change oxygen affinity (see Table S2), thus suggesting they are also involved in this allosteric equilibrium.”

Considering whether key residues can be identified by sequence analysis alone, it is interesting to provide a more general answer. As shown in Fig. 3, as well as in the equivalent Figs for others studied residues in other protein families, in many cases high SeqIC residues correlate with high FrustrIC value and an energetic interpretation can be made. However, many more residues also show high SeqIC values but not high FrustrIC (i.e. residues in highly dynamic or conformationally diverse regions). The opposite case also holds true, low

SeqIC residues can have high FrustrIC values (e.g. residues important for the hydrophobic core). Therefore, the combination of both FrustrIC indices, seems to be a much more selective method in terms of identifying relevant residues, compared to sequence conservation.

They then move on to analysing members of the 4 RAS sub families. The SRFI is not conserved between families, even in the 5 binding motifs. When treated separately the subfamilies reveal some highly frustrated sites – mainly connected to binding to the ligand. It is not clear to me exactly what the frustration analysis reveals over and above the SDPs and the identification of the binding site. Again metal-ion binding is highlighted – is this because the energetics do not include consideration of the ligands?

When an SDP analysis is performed, the only information that is retrieved from the algorithm is that there are certain positions in the MSA that change their predominant identity across families/clusters in the MSA. However, this analysis alone does not give any clue related to what each amino acid is doing at each cluster. With our analysis we can classify these residues as falling into one of two classes i.e. non-energetically conserved or energetically conserved. If energetically conserved, residues can be classified in any of the 3 frustration levels: minimally frustrated, neutral or highly frustrated which in turn allow to make more meaningful hypothesis about their role at each subfamily within the broader superfamily tree.

As explained before, the metal-ion even if present at the input structures is not parameterized in the hamiltonian and therefore is invisible to the algorithm. Because of this, the site where the metal-ion is supposed to bind is highly frustrated across all families. We have now briefly explained this in the text and a better explanation is also now present at the methods section where frustration is explained.

The Ras family part has been moved to a supplementary note while we left a summarized version of it in the main text. Further details have been included in the supplementary note. In particular we expanded the rationale behind moving from single residues to contacts to better retrieve the evolutionary signals. Moreover we have included a more detailed analysis of frustration in both SRFI and contacts results.

- The authors then look at the proteins of the SARS-COV2 virus from the perspective of frustration. They then included analysis of the Sarbecovirus and the PLpro domain of viral polyproteins, giving a detailed but inconclusive description of their observations.

We thank the reviewer for this comment. We have reanalysed this section and decided that the proteome wide analysis was deviating attention from our main goal with this section and decided to move it to a supplementary note. We have also included further details and thoughts into the PLPRo part that aims to communicate that unsupervised analysis of multiple evolutionary related families can shed some light into proteins that are not so well described or studied. We hope the reviewer finds it more concise and conclusive now.

- Lastly they have performed some ‘theoretical’ mutagenesis experiments, identifying ‘frustrated’ residues and mutating them to induce a change of ‘fold’ in RfaH. They tested

their prediction by running AlphaFold on the mutated sequences and inspecting the structures predicted.

In summary, this is a dense detailed paper, studying many families and subfamilies, from which it is quite challenging to draw conclusions, outside of specific observations for the specific families. I have read it several times and still struggle with the details and what insights can be drawn from using these frustration indices, that are not apparent from just using the sequences alone. The overall picture is still quite confusing. Perhaps some of the examples and detailed discussions of specific residues could be removed to give more clarity?

We really took note of these comment and because of that we have simplified the main text by moving some parts to supplementary notes so the main text is less dense and the supplementary material can also give space to further discuss those parts that although secondary, we believe are important so readers can have a broader context if needed. We hope the reviewer finds the new version of the article clearer and less dense.

This approach is another way to consider the problem of variation, that builds on recent advances in protein 3D structure prediction and how we can use structural data to provide additional insight. It is still challenging to identify accurately which variations are pathogenic – and I was surprised, given the scope of this paper, that the authors did not include this in their considerations. The results are complicated, vary between families, and are difficult to both understand and consider more broadly. I think the authors included too many examples, each of which requires biological understanding, and in so doing make the paper very difficult to read and comprehend. My main question remains ‘Is frustration conserved’ – one would have thought that unless absolutely needed, then the protein would evolve itself to a more ‘comfortable’ energetically favourable state. A clear statement on what the authors have found from their analyses would help.

We thank the reviewer for these last comments as we really think they helped us to think of a better way to convey our message. We have extended the message at the discussion section (text below) hoping that now the main question is addressed

“We have introduced the analysis of energetic patterns across protein families based on the conservation of frustration levels. These patterns reveal conserved physicochemical constraints related to protein stability and function, providing a biophysical interpretation of the impact of sequence divergence over evolutionary timescales. Proteins evolve constrained by the need of minimizing energetic conflicts related with folding and stability⁴² while paying an energetic cost to maintain functional sites^{2,43–45}. By collectively analyzing frustration in proteins with common ancestry, we have previously shown the presence of energetic constraints that exist to preserve stability and function in protein families. We observed the presence of mainly foldability constraints in the Ankyrin Repeat Protein family²¹ where consensus identities in the MSAs correlated with high stability signals. As a consequence of functional promiscuity and non-conserved interaction interfaces with their targets, there are no conserved and highly frustrated signals in the family. In contrast, highly frustrated interactions are found, mainly involving the catalytic residues, in globular cases such as the Beta Lactamases¹⁰. But beyond catalysis we also found non catalytic residues being highly frustrated that when mutated would have consequences on fitness and

antibiotic resistance. This suggested that frustration conservation analysis would have broader applications to the study of protein physiology.”

Reviewer #3 (Remarks to the Author):

In this manuscript, the authors integrate previously developed frustration calculation methods with new frustration conservation parameters to study the conservation of frustration in protein families and discuss the connection of such frustration with protein function. This is a new and interesting approach.

Comments:

(1) In order to make the manuscript broadly accessible (beyond bioinformaticians), the results section should be expanded and all terms need to not just be defined (in some places that has been omitted) but also explained.

We thank the reviewer for this observation. We have moved some parts of the results (RAS superfamily and SarsCov2) to supplementary material leaving a summarized version in the main text which gave us space so we can expand other parts. We hope the reviewer finds the article more accessible now.

(2) Similar to the above, frustration is still not commonly understood or used in the context of proteins or protein families. A few paragraphs explaining the term, how it is calculated and why frustration may occur in functional regions should be given with extensive references.

We thank the reviewer for noticing this in alignment with previous comments by other reviewers. We have expanded the frustration explanation in the methods section. We hope it is better explained for general readers now.

(3) The authors focus on stability and function but mention foldability only in passing. While in other protein families, foldability is a constraint, in fold-switchers, (and potentially other proteins that undergo conformational transitions), foldability is likely selected for. In this context, it would be useful to understand, what signals of foldability are likely to be seen by the frustration calculation methods used in the manuscript and what are likely missed. A larger discussion of foldability and frustration should also be given.

We thank the reviewer for their observation. We have introduced that foldability is also a variable to take into account at several points in the text where stability is mentioned. We also briefly discussed it in the RfaH section.

However, a more in depth discussion is not trivial as it is not possible to disentangle foldability from stability constraints from our analysis in the native states of protein family members. Yes, it is known that highly frustrated interactions related to function can affect foldability. Examples of this are the highly frustrated residues that are located within Im7 at the interaction site with colicin E7 that results in the appearance of a folding intermediate (Sutto et al, PNAS 2007, <https://doi.org/10.1073/pnas.0709922104>) and the consequent need of a chaperone that binds to those residues to prevent aggregation prone events (He et al, Science Advances 2016, DOI: 10.1126/sciadv.1601625). Another example, still not experimentally validated, was shown in Beta Lactamases where there is a highly frustrated Proline in the native state whose isomerization might constitute a relevant step for the foldability of the protein (Freiberger, PNAS et al 2019, <https://doi.org/10.1073/pnas.1819859116>). From just these specific examples, it seems that

some highly frustrated signals, related to functional aspects of proteins, negatively correlate with foldability and nature has found different strategies to account for them. However, it is not possible from our results on the native states of proteins to differentiate such events from other functional signals like protein-protein binding sites. Frustration studies over MD simulations could shed some light into foldability aspects, something we briefly explored (Rausch, et al Bioinformatics 2021, <https://doi.org/10.1093/bioinformatics/btab176>) but that is beyond the scope of the current article.

(4) Methods complementary to frustration analysis such as SCA and DCA (these are just examples) have been used to study protein families. These should be discussed, cited and contrasted with the frustration analysis.

We understand that DCA and SCA are important methods for protein structural prediction which is a different problem than the one we are focusing on. The couplings calculated by these methods predict physical proximity between pairs of residues in a protein or complex structure but are not predictive of functional regions. Therefore, we do not consider that a direct comparison between those methods and FrustraEvo would be meaningful for this article. There are some alternative DCA-based methods that can be used to predict the effect of a mutation in a given amino acid site within a protein (e.g. EVmutation from Debora Marks' Lab) which might be more suitable to detect the functional relevance of an amino acid site but still do not compare directly to evolutionary frustration analysis. In addition, the Wolynes group is working on a Frustratometer version that uses DCA derived energy functions, instead of the AWSEM one that is used in the current version. However, this methodology is under development and cannot be introduced yet as part of this work.

REVIEWERS' COMMENTS

Reviewer #1 (Remarks to the Author):

The authors have done an excellent job in addressing the comments of the reviewers, and indeed in relating to them in the Response to the referees. The concept of frustration, introduced by the Wolynes group is fundamental to the understanding of protein function, and the analysis carried out in this work on its conservation in families/superfamilies is an excellent idea, further emphasizing its functional significance.

I have no additional comments. This innovative paper can be accepted as is.

Reviewer #2 (Remarks to the Author):

The authors have made a very serious attempt to address all the queries put forward by the referees. They have also tried to simplify and streamline the paper to make it more comprehensible for the average reader. It still remains a complex discussion, but one worth publishing to stimulate consideration by others. Therefore I now recommend publication.

Reviewer #3 (Remarks to the Author):

I have no further comments about the science in the work.

Although it is mentioned briefly in the introduction, the authors should clearly state why residues that perform protein function might be frustrated. There are many reviews on this (<https://doi.org/10.1016/j.sbi.2019.11.005>; <https://doi.org/10.1016/j.sbi.2016.01.001>; <https://doi.org/10.1017/S0033583514000092>). The authors should expand their citations and clearly explain the tradeoffs between function and stability/foldability to make this manuscript more accessible to readers.

November 10th, Barcelona, Spain

After assessing the editorial and reviewers' comments we are very pleased to submit the response to reviewers as well as our revised manuscript.

We address here the comments from reviewer 3. Changes in the main text are coloured in red.

We are thankful to Reviewers 1 and 2 for their positive feedback on our revised manuscript.

REVIEWERS' COMMENTS

Reviewer #3 (Remarks to the Author):

I have no further comments about the science in the work.

Although it is mentioned briefly in the introduction, the authors should clearly state why residues that perform protein function might be frustrated. There are many reviews on this (<https://doi.org/10.1016/j.sbi.2019.11.005>; <https://doi.org/10.1016/j.sbi.2016.01.001>; <https://doi.org/10.1017/S0033583514000092>). The authors should expand their citations and clearly explain the tradeoffs between function and stability/foldability to make this manuscript more accessible to readers.

We thank the reviewer for the positive feedback on our revised manuscript. We have now stated why frustrated residues, related to function, might be frustrated.

Here is the text that we have included at the introduction in the main text. We have edited the first sentence and we have included what follows in bold, citing the reviews suggested by the referee.

These conflicting signals have been shown to be enriched around residues that are associated with many functional aspects of proteins such as the binding to small ligands or cofactors as well as protein-protein interactions⁸, allostery⁹, catalytic sites¹⁰, disorder/order transitions¹¹ or the existence of fuzzy regions¹². **Because amino acids at those positions are selected for functional reasons, they can often lead to more rugged energy landscapes, resulting in a trade-off between molecular function and local stability¹³. Many proteins have been shown to have reduced activity when stabilizing mutations are introduced at functional sites, showing the delicate equilibrium between stability and function and the functional importance of local frustration^{14,15}.**